# RETHINKING POSITIVE SAMPLING FOR CONTRASTIVE LEARNING WITH KERNEL

## ABSTRACT

Data augmentation is a crucial component in unsupervised contrastive learning (CL). It determines how positive samples are defined and, ultimately, the quality of the representation. Even if efforts have been made to find efficient augmentations for ImageNet, CL underperforms compared to supervised methods and it is still an open problem in other applications, such as medical imaging, or in datasets with easy-to-learn but irrelevant imaging features. In this work, we propose a new way to define positive samples using kernel theory along with a novel loss called *decoupled uniformity*. We propose to integrate prior information, learnt from generative models viewed as feature extractor, or given as auxiliary attributes, into contrastive learning, to make it less dependent on data augmentation. We draw a connection between contrastive learning and the conditional mean embedding theory to derive tight bounds on the downstream classification loss. In an unsupervised setting, we empirically demonstrate that CL benefits from generative models, such as VAE and GAN, to less rely on data augmentations. We validate our framework on vision and medical datasets including CIFAR10, CIFAR100, STL10, ImageNet100, CheXpert and a brain MRI dataset. In the weakly supervised setting, we demonstrate that our formulation provides state-of-the-art results.

## 1 INTRODUCTION

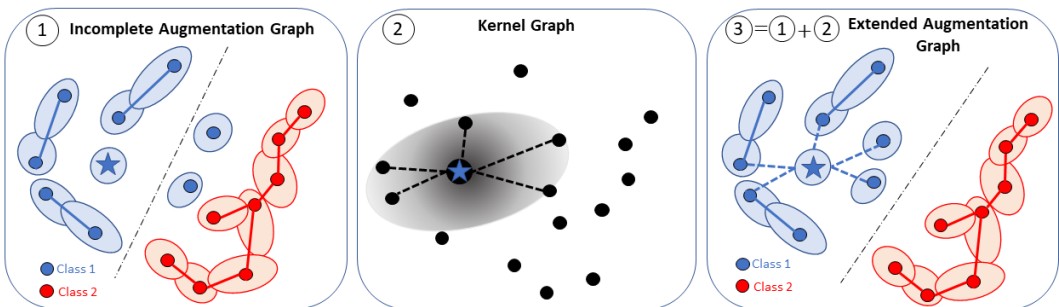

Figure 1: Illustration of the proposed method. Each point is an original image $\bar{x}$. Two points are connected if they can be transformed into the same augmented image using a distribution of augmentations $\mathcal{A}$. Colors represent semantic (unknown) classes and light disks represent the support of augmentations for each sample $\bar{x}$, $\mathcal{A}(\cdot|\bar{x})$. From an incomplete augmentation graph (1) where intra-class samples are not connected (e.g. augmentations are insufficient or not adapted), we reconnect them using a kernel defined on prior information (either learnt with generative model, viewed as feature extractor, or given as auxiliary attributes). The extended augmentation graph (3) is the union between the (incomplete) augmentation graph (1) and the kernel graph (2). In (2), the gray disk indicates the set of points $\bar{x}'$ that are close to the anchor (blue star) in the kernel space.

Contrastive Learning (CL)(44; 3; 4; 7; 10) is a paradigm designed for representation learning which has been applied to unsupervised (10; 13), weakly supervised(55; 20) and supervised problems (37). It gained popularity during the last years by achieving impressive results in the unsupervised

setting on standard vision datasets (*e.g.* ImageNet) where it almost matched the performance of its supervised counterpart (10; 29).

The objective in CL is to increase the similarity in the representation space between *positive* samples (semantically close), while decreasing the similarity between *negative* samples (semantically distinct). Despite its simple formulation, it requires the definition of a similarity function (that can be seen as an energy term (42)),and of a rule to decide whether a sample should be considered positive or negative. Similarity functions, such as the Euclidean scalar product (*e.g.* InfoNCE(44)), take as input the latent representations of an encoder $f \in \mathcal{F}$, such as a CNN (11) or a Transformer (9) for vision datasets.

In supervised learning (37), positives are simply images belonging to the same class while negatives are images belonging to different classes. In unsupervised learning (10), since labels are unknown, positives are usually defined as transformed versions (*views*) of the same original image (a.k.a. the anchor) and negatives are the transformed versions of all other images. As a result, the augmentation distribution $\mathcal{A}$ used to sample both positives and negatives is crucial (10) and it conditions the quality of the learnt representation. The most-used augmentations for visual representations involve aggressive crop and color distortion. Cropping induces representations with high occlusion invariance (46) while color distortion may avoids the encoder $f$ to take a shortcut (10) while aligning positive sample representations and fall into the simplicity bias (51).

Nevertheless, learning a representation that mainly relies on augmentations comes at a cost: both crop and color distortion induce strong biases in the final representation (46). Specifically, dominant objects inside images can prevent the model from learning features of smaller objects (12) (which is not apparent in object-centric datasets such as ImageNet) and few, irrelevant and easy-to-learn features, that are shared among views, are sufficient to collapse the representation (12) (a.k.a feature suppression). Finding the right augmentations in other visual domains, such as medical imaging, remains an open challenge (20) since we need to find transformations that preserve semantic anatomical structures (e.g. discriminative between pathological and healthy) while removing unwanted noise. If the augmentations are too weak or inadequate to remove irrelevant signal w.r.t. a discrimination task, then how can we define positive samples?

In our work, we propose to integrate *prior information*, learnt from generative models or given as auxiliary attributes, into contrastive learning, to make it less dependent on data augmentation. Using the theoretical understanding of CL through the augmentation graph, we make the connection with kernel theory and introduce a novel loss with theoretical guarantees on downstream performance. Prior information is integrated into the proposed contrastive loss using a kernel. In the unsupervised setting, we leverage pre-trained generative models, such as GAN (24) and VAE (38), to learn *a prior representation* of the data. We provide a solution to the feature suppression issue in CL (12) and also demonstrate SOTA results with weaker augmentations on visual benchmarks. In visual domain where data augmentations are not adapted to the downstream task (e.g. medical imaging), we show that we can improve CL, alleviating the need to find efficient augmentations. In the weakly supervised setting, we use instead auxiliary/prior information, such as image attributes (e.g. birds color or size) and we show better performance than previous conditional formulations based on these attributes (55).

In summary, we make the following contributions:

1. We propose a new framework for contrastive learning allowing the integration of prior information, learnt from generative models or given as auxiliary attributes, into the positive sampling.
2. We derive theoretical bounds on the downstream classification risk that rely on weaker assumptions for data augmentations than previous works on CL.
3. We empirically show that our framework can benefit from the latest advances of generative models to learn a better representation while relying on less augmentations.
4. We show that we achieve SOTA results in the unsupervised and weakly supervised setting.

## 2 RELATED WORKS

In a weakly supervised setting, recent studies (20; 55) have shown that positive samples can be defined conditionally to an auxiliary attribute in order to improve the final representation, in particular for medical imaging (20). From an information bottleneck perspective, these approaches essentially

compress the representation to be predictive of the auxiliary attributes. This might harm the performance of the model when these attributes are too noisy to accurately approximate the true semantic labels for a given downstream task.

In an unsupervised setting, recent approaches (22; 65; 66; 43) used the encoder $f$, learnt during optimization, to extend the positive sampling procedure to other views of different instances (*i.e.* distinct from the anchor) that are close to the anchor in the latent space. In order to avoid representation collapse, multiple instances of the same sample (2), a support set (22), a momentum encoder (43) or another small network (65) can be used to select the positive samples. In clustering approaches (43; 8), distinct instances with close semantics are attracted in the latent space using prototypes. These prototypes can be estimated through K-means (43) or Sinkhorn-Knopp algorithm (8). All these methods rely on the past representation of a network to improve the current one. They require strong augmentations and they essentially assume that the closest points in the representation space belong to the same latent class in order to better select the positives. This inductive bias is still poorly understood from a theoretical point of view (50) and may depend on the visual domain. For medical imaging, ImageNet self-supervised pre-training was beneficial for all subsequent tasks (2).

Our work also relates to generative models for learning representations. VAEs (38) learn the data distribution by mapping each input to a Gaussian distribution that we can easily sample from to reconstruct the original image. GANs (24), instead, sample directly from a Gaussian distribution to generate images that are classified by a discriminator in a min-max game. The discriminator representation can then be used (48) as feature extractor. Other models (ALI (21), BiGAN (18) and BigBiGAN(19)) learn simultaneously a generator and an encoder that can be used directly for representation learning (63). All these models do not require particular augmentations to model the data distribution but they perform generally poorer than recent discriminative approaches (64; 11) for representation learning. A first connection between generative models and contrastive learning has emerged very recently (36). In (36), authors study the feasibility of learning effective visual representations using only generated samples, and not real ones, with a contrastive loss. Their empirical analysis is complementary to our work. Here, we leverage the representation capacity of the generative models, rather than their generative power, to learn prior representation of the data.

## 3    CONSTRASTIVE LEARNING WITH DECOUPLED UNIFORMITY

**Problem setup.** The general problem in contrastive learning is to learn a data representation using an encoder $f \in \mathcal{F} : \mathcal{X} \to \mathbb{S}^{d-1}$ that is pre-trained with a set of $n$ original samples $(\bar{x}_i)_{i \in [1..n]} \in \bar{\mathcal{X}}$, sampled from the data distribution $p(\bar{x})$[1] These samples are transformed to generate *positive samples* (*i.e.*, semantically similar to $\bar{x}$) in $\mathcal{X}$, space of augmented images, using a distribution of augmentations $\mathcal{A}(\cdot|\bar{x})$. Concretely, for each $\bar{x}_i$, we can sample views of $\bar{x}_i$ using $x \sim \mathcal{A}(\cdot|\bar{x}_i)$ (*e.g.*, by applying color jittering, flip or crop with a given probability). For consistency, we assume $\mathcal{A}(\bar{x}) = p(\bar{x})$ so that the distributions $\mathcal{A}(\cdot|\bar{x})$ and $p(\bar{x})$ induce a marginal distribution $p(x)$ over $\mathcal{X}$. Given an anchor $\bar{x}_i$, all views $x \sim \mathcal{A}(\cdot|\bar{x}_j)$ from different samples $\bar{x}_{j \neq i}$ are considered as *negatives*. Once pre-trained, the encoder $f$ is fixed and its representation $f(\bar{\mathcal{X}})$ is evaluated through linear evaluation on a classification task using a labeled dataset $\mathcal{D} = \{(\bar{x}_i, y_i)\} \in \bar{\mathcal{X}} \times \mathcal{Y}$ where $\mathcal{Y} = [1..K]$, with $K$ the number of classes.

**Linear evaluation.** To evaluate the representation of $f$ on a classification task, we train a linear classifier $g(\bar{x}) = W f(\bar{x})$ ($f$ is fixed) that minimizes the multi-class classification error.

**Objective.** The popular InfoNCE loss (45; 44), often used in CL, imposes 1) alignment between positives and 2) uniformity between the views ($x \overset{\text{i.i.d.}}{\sim} \mathcal{A}(\cdot|\bar{x})$) of all instances $\bar{x}$ (57)– two properties that correlate well with downstream performance. However, by imposing uniformity between *all* views, we essentially try to both attract (alignment) and repel (uniformity) positive samples and therefore we cannot achieve a perfect alignment *and* uniformity, as noted in (57). Moreover, InfoNCE has been originally designed for only two views (i.e., one couple of positive) and its extension to multiple views is not straightforward (40). Previous works have proposed a solution to either the first (54) or second (60) issue. Here, we propose a modified version of the uniformity loss, presented in (57), that solves both issues since it: i) decouples positives from negatives, similarly to (60) and ii) is

---

[1]With an abuse of notation, we define it as $p(\bar{x})$ instead than $p_{\bar{X}}$ to simplify the presentation, as it is common in the literature

generalizable to multi-views as in (54). We introduce the Decoupled Uniformity loss as:

$$\mathcal{L}_{unif}^d(f) = \log \mathbb{E}_{p(\bar{x})p(\bar{x}')} e^{-||\mu_{\bar{x}} - \mu_{\bar{x}'}||^2} \tag{1}$$

where $\mu_{\bar{x}} = \mathbb{E}_{\mathcal{A}(x|\bar{x})} f(x)$ is called a *centroid* of the views of $\bar{x}$. This loss essentially repels distinct centroids $\mu_{\bar{x}}$ through an average pairwise Gaussian potential. Interestingly, it implicitly optimizes alignment between positives through the maximization of $||\mu_{\bar{x}}||^2$, so we do not need to explicitly add an alignment term. It can be shown (see Appendix B), that minimizing this loss brings to a representation space where the sum of similarities between views of the same sample is greater than the sum of similarities between views of different samples.

We will study its main properties hereafter and we will see that, contrary to other contrastive losses, *prior* information can be added during the estimation step of these centroids using a kernel. First, we define a measure of the risk on a downstream task.

**Supervised risk.** While previous analysis (58; 1) generally used the mean cross-entropy loss (as it has closer analytic form with InfoNCE), we use a supervised loss closer to decoupled uniformity with the same guarantees as the mean cross-entropy loss (see Appendix C.1). Notably, the geometry of the representation space at optimum is the same as cross-entropy and SupCon (37) and we can theoretically achieve perfect linear classification.

**Definition 3.1.** (Downstream supervised loss) For a given downstream task $\mathcal{D} = \bar{\mathcal{X}} \times \mathcal{Y}$, we define the classification loss as: $\mathcal{L}_{sup}(f) = \log \mathbb{E}_{y,y' \sim p(y)p(y')} e^{-||\mu_y - \mu_{y'}||^2}$, where $\mu_y = \mathbb{E}_{p(\bar{x}|y)} \mu_{\bar{x}}$.

This loss depends on centroids $\mu_{\bar{x}}$ rather than $f(\bar{x})$. Empirically, it has been shown (23) that performing feature averaging gives better performance on the downstream task.

## 3.1 GEOMETRICAL ANALYSIS OF DECOUPLED UNIFORMITY

**Definition 3.2.** (Finite-samples estimator) For $n$ samples $(\bar{x}_i)_{i \in [1..n]} \overset{i.i.d.}{\sim} p(\bar{x})$, the (biased) empirical estimator of $\mathcal{L}_{unif}^d(f)$ is: $\hat{\mathcal{L}}_{unif}^d(f) = \log \frac{1}{n(n-1)} \sum_{i \neq j} e^{-||\mu_{\bar{x}_i} - \mu_{\bar{x}_j}||^2}$. It converges in law to $\mathcal{L}_{unif}^d(f)$ with rate $O(n^{-1/2})$. Proof in Appendix E.1

**Theorem 1.** (Optimality of Decoupled Uniformity) Given $n$ points $(\bar{x}_i)_{i \in [1..n]}$ such that $n \leq d+1$, any optimal encoder $f^*$ minimizing $\hat{\mathcal{L}}_{unif}^d$ achieves a representation s.t.:
1. (Perfect uniformity) All centroids $(\mu_{\bar{x}_i})_{i \in [1..n]}$ make a regular simplex on the hyper-sphere $\mathbb{S}^{d-1}$
2.(Perfect alignment) $f^*$ is perfectly aligned, i.e $\forall x, x' \sim \mathcal{A}(\cdot|\bar{x}_i), f^*(x) = f^*(x')$ for all $i \in [1..n]$.
Proof in Appendix E.2.

Theorem 1 gives a complete geometrical characterization when the batch size $n$ set during training is not too large compared to the representation space dimension $d$. By removing the coupling between positives and negatives, we see that Decoupled Uniformity can realize both perfect alignment and uniformity, contrary to InfoNCE (57).

Most recent theories about CL (58; 28) make the hypothesis that samples from the same semantic class have overlapping augmented views to provide guarantees on the downstream task when optimizing InfoNCE (10) or Spectral Contrastive loss (28). This assumption, known as intra-class connectivity hypothesis, is very strong and only relies on the augmentation distribution $\mathcal{A}$. In particular, augmentations should not be "too weak", so that all intra-class samples are connected among them, and at the same time not "too strong", to prevent connections between inter-class samples and thus preserve the semantic information. Here, we prove that we can relax this hypothesis if we can provide a kernel (viewed as a similarity function between original samples $\bar{x}$) that is "good enough" to relate intra-class samples not connected by the augmentations (see Fig. 1). In practice, we show that generative models (viewed as feature extractor) or auxiliary information can define such kernel. We first recall the definition of the augmentation graph (58), and intra-class connectivity hypothesis before presenting our main theorems. For simplicity, we assume that the set of images $\bar{\mathcal{X}}$ is finite (similarly to (58; 28)). Our bounds and theoretical guarantees will never depend on the cardinality $|\bar{\mathcal{X}}|$.

---

[2] By Jensen's inequality $||\mu_{\bar{x}}|| \leq \mathbb{E}_{\mathcal{A}(x|\bar{x})} ||f(x)|| = 1$ with equality iff $f$ is constant on $\text{supp}\, \mathcal{A}(\cdot|\bar{x})$.

**INTRA-CLASS CONNECTIVITY HYPOTHESIS.**

**Definition 3.3.** (Augmentation graph (28; 58)) Given a set of original images $\bar{\mathcal{X}}$, we define the augmentation graph $G_{\mathcal{A}}(V, E)$ for an augmentation distribution $\mathcal{A}$ through 1) a set of vertices $V = \bar{\mathcal{X}}$ and 2) a set of edges $E$ such that $(\bar{x}, \bar{x}') = e \in E$ if the two original images $\bar{x}, \bar{x}'$ can be transformed into the same augmented image through $\mathcal{A}$, i.e $\operatorname{supp} \mathcal{A}(\cdot|\bar{x}) \cap \operatorname{supp} \mathcal{A}(\cdot|\bar{x}') \neq \emptyset$.

Previous analysis in CL make the hypothesis that there exists an optimal (accessible) augmentation module $\mathcal{A}^*$ that fulfills:

**Assumption 1.** (Intra-class connectivity (58)) For a given downstream classification task $\mathcal{D} = \bar{\mathcal{X}} \times \mathcal{Y}$ $\forall y \in \mathcal{Y}$, the augmentation subgraph, $G_y \subset G_{\mathcal{A}^*}$ containing images only from class $y$, is connected.

Under this hypothesis, Decoupled Uniformity loss can also tightly bound the downstream supervised risk *but* for a bigger class of encoders than prior work (not restricted to L-smooth functions (58)).

**Definition 3.4.** (Weak-aligned encoder) An encoder $f \in \mathcal{F}$ is $\epsilon'$-weak ($\epsilon' \geq 0$) aligned on $\mathcal{A}$ if:

$$||f(x) - f(x')|| \leq \epsilon' \qquad \forall \bar{x} \in \bar{\mathcal{X}}, \forall x, x' \overset{i.i.d.}{\sim} \mathcal{A}(\cdot|\bar{x})$$

**Theorem 2.** (Guarantees with $\mathcal{A}^*$) Given an optimal augmentation module $\mathcal{A}^*$, for any $\epsilon$-weak aligned encoder $f \in \mathcal{F}$ we obtain: $\mathcal{L}_{unif}^d(f) \leq \mathcal{L}_{sup}(f) \leq 8D\epsilon + \mathcal{L}_{unif}^d(f)$ where $D$ is the maximum diameter of all intra-class graphs $G_y$ ($y \in \mathcal{Y}$). Proof in Appendix E.5.

Contrary to previous work (58), this theorem does not require L-smoothness of $f \in \mathcal{F}$ (strong assumption) and provides tighter lower bound. In practice, the diameter $D$ can be controlled by a small constant in some cases (*e.g.,* 4 in (58)) but it remains specific to the dataset at hand. Furthermore, we observe (see Appendix A.1) that $f$ realizes alignment with small error $\epsilon$ during optimization of $\mathcal{L}_{unif}^d(f)$ for augmentations close to the sweet spot $\mathcal{A}^*$ (54) on CIFAR-10 and CIFAR-100.

In the next section, we study the case when $\mathcal{A}^*$ is not accessible or very hard to find.

### 3.2 RECONNECT THE DISCONNECTED: EXTENDING THE AUGMENTATION GRAPH WITH KERNEL

Having access to optimal augmentations is a strong assumption and, for many real-world applications (e.g medical imaging (20)), it may not be accessible. If we have only weak augmentations (*e.g.*, $\operatorname{supp} \mathcal{A}(\cdot|\bar{x}) \subsetneq \operatorname{supp} \mathcal{A}^*(\cdot|\bar{x})$ for any $\bar{x}$), then some intra-class points might not be connected and we would need to reconnect them to ensure good downstream accuracy (see Theorem 7 in Appendix C.2). Augmentations are intuitive and they have been hand-crafted for decades by using human perception (*e.g.*, a rotated chair remains a chair and a gray-scale dog is still a dog). However, we may know other *prior information* about objects that are difficult to transfer through invariance to augmentations (*e.g.*, chairs should have 4 legs). This prior information can be either given as image attributes (*e.g.*, age or sex of a person, color of a bird, etc.) or, in an unsupervised setting, directly learnt through a generative model (*e.g.*, GAN or VAE). Now, we ask: how can we integrate this information inside a contrastive framework to reconnect intra-class images that are actually disconnected in $G_{\mathcal{A}}$? We rely on conditional mean embedding theory and use a kernel defined on the prior representation/information. This allows us to estimate a better configuration of the centroids in the representation space, with respect to the downstream task, and, ultimately, provide theoretical guarantees on the classification risk.

**$\epsilon$-KERNEL GRAPH.**

**Definition 3.5.** (RKHS on $\bar{\mathcal{X}}$) We define the RKHS $(\mathcal{H}_{\bar{\mathcal{X}}}, K_{\bar{\mathcal{X}}})$ on $\bar{\mathcal{X}}$ associated with a kernel $K_{\bar{\mathcal{X}}}$.

**Example.** If we work with large natural images, assuming that we know a prior $z(\bar{x})$ about our images (*e.g.*, given by a generative model), we can compute $K_{\bar{\mathcal{X}}}$ using $z$ through $K_{\bar{\mathcal{X}}}(\bar{x}, \bar{x}') = \tilde{K}(z(\bar{x}), z(\bar{x}'))$ where $\tilde{K}$ is a standard kernel (*e.g.*, Gaussian or Cosine).

To link kernel theory with the previous augmentation graph, we need to define a *kernel graph* that connects images with high similarity in the kernel space.

**Definition 3.6.** ($\epsilon$-Kernel graph) Let $\epsilon > 0$. We define the $\epsilon$-kernel graph $G_{K_{\bar{\mathcal{X}}}}^{\epsilon}(V, E_K)$ for the kernel $K_{\bar{\mathcal{X}}}$ on $\bar{\mathcal{X}}$ through 1) a set of vertices $V = \bar{\mathcal{X}}$ and 2) a set of edges $E_{K_{\bar{\mathcal{X}}}}$ such that $e \in E_{K_{\bar{\mathcal{X}}}}$ between $\bar{x}, \bar{x}' \in \bar{\mathcal{X}}$ iff $\max(K_{\bar{\mathcal{X}}}(\bar{x}, \bar{x}), K_{\bar{\mathcal{X}}}(\bar{x}', \bar{x}')) - K_{\bar{\mathcal{X}}}(\bar{x}, \bar{x}') \leq \epsilon$.

The condition $\max(K_{\bar{\mathcal{X}}}(\bar{x}, \bar{x}), K_{\bar{\mathcal{X}}}(\bar{x}', \bar{x}')) - K_{\bar{\mathcal{X}}}(\bar{x}, \bar{x}') \leq \epsilon$ implies that $d_{K_{\bar{\mathcal{X}}}}(\bar{x}, \bar{x}') \leq 2\epsilon$ where $d_{K_{\bar{\mathcal{X}}}}(\bar{x}, \bar{x}') = K_{\bar{\mathcal{X}}}(\bar{x}, \bar{x}) + K_{\bar{\mathcal{X}}}(\bar{x}', \bar{x}') - 2K_{\bar{\mathcal{X}}}(\bar{x}, \bar{x}')$ is the kernel distance. For kernels with constant norm (*e.g.*, the standard Gaussian, Cosine or Laplacian kernel), it is in fact an equivalence. Intuitively, it means that we connect two original points in the kernel graph if they have small distance in the kernel space. We give now our main assumption to derive a better estimator of the centroid $\mu_{\bar{x}}$ in the insufficient augmentation regime.

**Assumption 2.** (Extended intra-class connectivity) For a given task $\mathcal{D} = \bar{\mathcal{X}} \times \mathcal{Y}$, the extended graph $\tilde{G} = G_{\mathcal{A}} \cup G_{K_{\bar{\mathcal{X}}}}^{\epsilon} = (V, E \cup E_{K_{\bar{\mathcal{X}}}})$ (union between augmentation graph and $\epsilon$-kernel graph) is class-connected for all $y \in \mathcal{Y}$.

This assumption is notably weaker than Assumption 1 w.r.t augmentation distribution $\mathcal{A}$. Here, we do not need to find the optimal distribution of augmentations $\mathcal{A}^*$, as long as we have a kernel $K_{\bar{\mathcal{X}}}$ such that disconnected points in the augmentation graph are connected in the $\epsilon$-kernel graph. If $K$ is not well adapted to the data-set (i.e it gives very low values for intra-class points), then $\epsilon$ needs to be large to re-connect these points and, as shown in Appendix A.2, the classification error will be high. In practice, this means that we need to tune the hyper-parameter of the kernel (e.g., $\sigma$ for a RBF kernel) so that all intra-class points are reconnected with a small $\epsilon$.

#### CONDITIONAL MEAN EMBEDDING.

Decoupled Uniformity loss includes no kernel in its raw form. It only depends on centroids $\mu_{\bar{x}} = \mathbb{E}_{\mathcal{A}(x|\bar{x})} f(x)$. Here, we show that another consistent estimator of these centroids can be defined, using the previous kernel $K_{\bar{\mathcal{X}}}$. To show it, we **fix** an encoder $f \in \mathcal{F}$ and require the following technical assumption in order to apply conditional mean embedding theory (52; 39).

**Assumption 3.** (Expressivity of $K_{\bar{\mathcal{X}}}$) The (unique) RKHS $(\mathcal{H}_f, K_f)$ defined on $\mathcal{X}$ with kernel $K_f = \langle f(\cdot), f(\cdot) \rangle_{\mathbb{R}^d}$ fulfills $\forall g \in \mathcal{H}_f, \mathbb{E}_{\mathcal{A}(x|\cdot)} g(x) \in \mathcal{H}_{\bar{\mathcal{X}}}$

**Theorem 3.** (Centroid estimation) Let $(x_i, \bar{x}_i)_{i \in [1..n]} \overset{iid}{\sim} \mathcal{A}(x, \bar{x})$. Assuming 3, a consistent estimator of the centroid is:

$$\forall \bar{x} \in \bar{\mathcal{X}}, \hat{\mu}_{\bar{x}} = \sum_{i=1}^{n} \alpha_i(\bar{x}) f(x_i) \tag{2}$$

where $\alpha_i(\bar{x}) = \sum_{j=1}^{n} [(K_n + n\lambda \mathbf{I}_n)^{-1}]_{ij} K_{\bar{\mathcal{X}}}(\bar{x}_j, \bar{x})$ and $K_n = [K_{\bar{\mathcal{X}}}(\bar{x}_i, \bar{x}_j)]_{i,j \in [1..n]}$. It converges to $\mu_{\bar{x}}$ with the $\ell_2$ norm at a rate $O(n^{-1/4})$ for $\lambda = O(n^{-1/2})$. Proof in Appendix E.6.

**Intuition.** This theorem says that we can use representations of images close to an anchor $\bar{x}$, according to our prior information, to accurately estimate $\mu_{\bar{x}}$. Consequently, if the prior is "good enough" to connect intra-class images disconnected in the augmentation graph (i.e. fulfills Assumption 2), then this estimator allows us to tightly control the classification risk. From this theorem, we naturally derive the empirical Kernel Decoupled Uniformity loss using the previous estimator.

**Definition 3.7.** (Empirical Kernel Decoupled Uniformity Loss) Let $(x_i, \bar{x}_i)_{i \in [1..n]} \overset{iid}{\sim} \mathcal{A}(x, \bar{x})$. Let $\hat{\mu}_{\bar{x}_j} = \sum_{i=1}^{n} \alpha_{i,j} f(x_i)$ with $\alpha_{i,j} = ((K_n + \lambda n \mathbf{I}_n)^{-1} K_n)_{ij}$, $\lambda = O(n^{-1/2})$ a regularization constant and $K_n = [K_{\bar{\mathcal{X}}}(\bar{x}_i, \bar{x}_j)]_{i,j \in [1..n]}$. We define the empirical kernel decoupled uniformity loss as:

$$\hat{\mathcal{L}}_{unif}^{d}(f) \overset{\text{def}}{=} \log \frac{1}{n(n-1)} \sum_{i,j=1}^{n} \exp(-||\hat{\mu}_{\bar{x}_i} - \hat{\mu}_{\bar{x}_j}||^2) \tag{3}$$

**Extension to multi-views.** If we have $V$ views $(x_i^{(v)})_{v \in [1..V]}$ for each $\bar{x}_i$, we can easily extend the previous estimator with $\hat{\mu}_{\bar{x}_i} = \frac{1}{V} \sum_{v=1}^{V} \hat{\mu}_{\bar{x}_j}^{(v)}$ where $\hat{\mu}_{\bar{x}_j}^{(v)} = \sum_{i=1}^{n} \alpha_{i,j} f(x_i^{(v)})$.

The computational cost added is roughly $O(n^3)$ (to compute the inverse matrix of size $n \times n$) but it remains negligible compared to the back-propagation time using classical stochastic gradient descent. Importantly, the gradients associated to $\alpha_{i,j}$ are not computed.

#### A TIGHT BOUND ON THE CLASSIFICATION LOSS WITH WEAKER ASSUMPTIONS.

We show here that $\hat{\mathcal{L}}_{unif}^{d}(f)$ can tightly bound the supervised classification risk for well-aligned encoders $f \in \mathcal{F}$.

**Theorem 4.** We assume 2 and 3 hold for a reproducible kernel $K_{\bar{\mathcal{X}}}$ and augmentation distribution $\mathcal{A}$. Let $(x_i, \bar{x}_i)_{i \in [1..n]} \overset{iid}{\sim} \mathcal{A}(x, \bar{x})$. For any $\epsilon'$-weak aligned encoder $f \in \mathcal{F}$:

$$\hat{\mathcal{L}}^d_{unif}(f) - O\left(n^{-1/4}\right) \leq \mathcal{L}_{sup}(f) \leq \hat{\mathcal{L}}^d_{unif}(f) + 4D(2\epsilon' + \beta_n(K_{\bar{\mathcal{X}}})\epsilon) + O\left(n^{-1/4}\right) \quad (4)$$

where $\beta_n(K_{\bar{\mathcal{X}}}) = (\frac{\lambda_{min}(K_n)}{\sqrt{n}} + \sqrt{n}\lambda)^{-1} = O(1)$ for $\lambda = O(n^{-1/2})$, $K_n = (K_{\bar{\mathcal{X}}}(\bar{x}_i, \bar{x}_j))_{i,j \in [1..n]}$ and $D$ is the maximal diameter of all sub-graphs $\tilde{G}_y \subset \tilde{G}$ where $y \in \mathcal{Y}$. We noted $\lambda_{min}(K_n) > 0$ the minimal eigenvalue of $K_n$. Proof in Appendix E.7.

**Interpretation.** Theorem 4 gives tight bounds on the classification loss $\mathcal{L}_{sup}(f)$ with weaker assumptions than current work (1; 58; 28). We don't require perfect alignment for $f \in \mathcal{F}$ or L-smoothness and we don't have class collision term (even if the extended augmentation graph may contain edges between inter-class samples), contrarily to (1). Also, the estimation error doesn't depend on the number of views (which is low in practice))–as it was always the case in previous formulations (58; 1; 28) – but rather on the batch size $n$ and the eigenvalues of the kernel matrix (controlling the variance of the centroid estimator (27)) . Contrarily to CCLK (55), we don't condition our representation to weak attributes but rather we provide better estimation of the conditional mean embedding conditionally to the original image. Eventually, our loss remains in an unconditional contrastive framework driven by the augmentations $\mathcal{A}$ and the prior $K_{\bar{\mathcal{X}}}$ on input images. Theorem 2 becomes a special case $\epsilon = 0$ and $\tilde{\mathcal{A}} = \mathcal{A}^*$ (i.e the augmentation graph is class-connected, a stronger assumption than 2). In Appendix A.2, we provide empirical evidence that better kernel quality (measured by k-NN accuracy in kernel graph) improves downstream accuracy, as theoretically expected by the theorem. It also provides a new way to select *a priori* a good kernel.

## 4 EXPERIMENTS

Here, we study several problems where Kernel Decoupled Uniformity outperforms current contrastive SOTA models. In unsupervised learning, we show that we can leverage generative models representation to outperform current self-supervised models when the augmentations are insufficient to remove irrelevant signals from images. In weakly supervised, we demonstrate the superiority of our unconditional formulation when noisy auxiliary attributes are available. Implementation details are presented in Appendix D. We systematically use 2 views for fairness in our main experiments.

| Model | 0 bits | 5 bits | 10 bits | 20 bits |
|---|---|---|---|---|
| SimCLR (10) | 79.4 | 68.74 | 13.67 | 10.07 |
| BYOL (26) | 80.14 | 19.98 | 10.33 | 10.00 |
| $\beta$-VAE ($\beta = 1$) | 41.37 | 43.32 | 42.94 | 43.1 |
| $\beta$-VAE ($\beta = 2$) | 42.28 | 43.89 | 43.11 | 42.19 |
| $\beta$-VAE ($\beta = 4$) | 42.5 | 42.5 | 42.5 | 39.87 |
| Decoupled Unif (ours) | 82.43 | 53.45 | 10.08 | 9.64 |
| $K_{VAE}$ Decoupled Unif (ours) | **82.74**$_{\pm 0.18}$ | **68.75**$_{\pm 0.24}$ | **68.42**$_{\pm 0.51}$ | **68.58**$_{\pm 0.17}$ |

Table 1: Linear evaluation accuracy (%) on RandBits-CIFAR10 with ResNet18 for 200 epochs. For VAE, we use a ResNet18 backbone. Once trained, we use its representation to define the kernel $K_{VAE}$ in kernel decoupled uniformity loss.

**Evading feature suppression with VAE.** Previous investigations (12) have shown that a few easy-to-learn irrelevant features not removed by augmentations can prevent the model from learning all semantic features inside images. We propose here a first solution to this issue.

RandBits dataset (12). We build a RandBits dataset based on CIFAR-10. For each image, we add a random integer sampled in $[0, 2^k - 1]$ where $k$ is a controllable number of bits. To make it easy to learn, we take its binary representation and repeat it to define $k$ channels that are added to the original RGB channels. Importantly, these channels will not be altered by augmentations, so they will be shared across views. We train a ResNet18 on this dataset with standard SimCLR augmentations (10) and varying $k$. For kernel decoupled uniformity, we use a $\beta$-VAE representation (ResNet18 backbone, $\beta = 1$, also trained on RandBits) to define $K_{VAE}(\bar{x}, \bar{x}') = K(\mu(\bar{x}), \mu(\bar{x}'))$

where $\mu(\cdot)$ is the mean Gaussian distribution of $\bar{x}$ in the VAE latent space and $K$ is a standard RBF kernel.

Table 1 shows the linear evaluation accuracy computed on a fixed encoder trained with various contrastive (SimCLR, Decoupled Uniformity and Kernel Decoupled Uniformity) and non-contrastive (BYOL and $\beta$-VAE) methods. As noted previously (12), $\beta$-VAE is the only method insensitive to the number of added bits, but its representation quality remains low compared to other discriminative approaches. All contrastive approaches fail for $k \geq 10$ bits. This can be explained by noticing that, as the number of bits $k$ increases, the number of edges between intra-class images in the augmentation graph $G_{\mathcal{A}}$ decreases. For $k$ bits, on average $N/2^k$ images share the same random bits ($N = 50000$ is the dataset size). So only these images can be connected in $G_{\mathcal{A}}$. For $k = 20$ bits, $< 1$ image share the same bits which means that they are almost all disconnected, and it explains why standard contrastive approaches fail. Same trend is observed for non-contrastive approaches (*e.g.* BYOL) with a degradation in performance even faster than SimCLR. Interestingly, encouraging a disentangled representation by imposing higher $\beta > 1$ in $\beta$-VAE does not help. Only our $K_{VAE}$ Decoupled Uniformity loss obtains good scores, regardless of the number of bits.

**BigBiGAN as prior.** We show that very recent advances in generative modeling improve representations of contrastive models in Table 2 with our approach. Due to our limited computational resources, we study ImageNet100 (54) (100-class subset of ImageNet used in the literature (54; 15; 57)) and we leverage BigBiGAN representation (19) as prior. In particular, we use BigBiGAN pre-trained on ImageNet to define a kernel $K_{GAN}(\bar{x}, \bar{x}') = K(z(\bar{x}), z(\bar{x}'))$ (with $K$ an RBF kernel and $z(\cdot)$ BigBiGAN's encoder). We demonstrate SOTA representation with this prior compared to all other contrastive and non-contrastive approaches. We use ResNet50 trained for 400 epochs. Please note that in our implementation, we do not use data augmentation for testing (i.e. during linear evaluation).

| Model | ImageNet100 |
|---|---|
| SimCLR (10) | 66.52 |
| BYOL (26) | 72.26 |
| CMC* (54) | 73.58 |
| DCL* (15) | 74.6 |
| AlignUnif* (57) | 74.6 |
| IFM (49) | 71.88 |
| BigBiGAN (19) | 72.0 |
| Decoupled Unif | 72.24 |
| $K_{GAN}$ Decoupled Unif | **76.58** |
| Supervised | $82.1_{\pm 0.59}$ |

Table 2: Linear evaluation accuracy (%) on ImageNet100. *Results from papers

| Model | CIFAR-10 | | | CIFAR-100 | | | STL-10 | | |
|---|---|---|---|---|---|---|---|---|---|
| | *All* | w/o Color | w/o Color and Crop | *All* | w/o Color | w/o Color and Crop | *All* | w/o Color | w/o Color and Crop |
| SimCLR (10) | 81.75 | 62.56 | 34.07 | 53.02 | 38.27 | 15.28 | 79.02 | 59.01 | 39.56 |
| BYOL (26) | 81.97 | 64.86 | 45.88 | 53.65 | 35.61 | 22.48 | 79.61 | 65.36 | 11.28 |
| MoCo v3 (14) | **86.51** | 67.71 | 42.12 | 58.83 | 36.95 | 22.11 | **83.02** | 64.25 | 38.38 |
| VAE* (38) | 41.37 | 41.37 | 41.37 | 14.34 | 14.34 | 14.34 | 42.17 | 42.17 | 42.17 |
| DCGAN* (48) | 66.71 | 66.71 | 66.71 | 26.17 | 26.17 | 26.17 | 70.06 | 70.06 | **70.06** |
| Decoupled Unif | 85.82 | 60.45 | 39.18 | **58.89** | 34.16 | 14.58 | 79.89 | 54.53 | 36.81 |
| $K_{VAE}$ Decoupled Unif | 85.84 | 72.92 | 50.52 | 58.19 | 45.59 | 28.24 | 79.10 | 61.39 | 45.64 |
| $K_{GAN}$ Decoupled Unif | 85.85 | **77.16** | **69.19** | 58.42 | **50.07** | **35.98** | 79.97 | **71.44** | 68.11 |

Table 3: When augmentation overlap hypothesis is not fulfilled, generative models can provide a good kernel to connect intra-class points not connected by augmentations. * For VAE and DCGAN, no augmentations were used during training. Bold: best result; underlined: second best. All models have been trained for 400 epochs.

**Towards weaker augmentations.** Color distortion (including color jittering and gray-scale) and crop are the two most important augmentations for SimCLR and other contrastive models to ensure a good representation on ImageNet (10). Whether they are best suited for other datasets (e.g medical imaging (20) or multi-objects images (12)) is still an open question. Here, we ask: can generative models remove the need for such strong augmentations? We use standard benchmarking datasets (CIFAR-10, CIFAR-100 and STL-10) and we study the case where augmentations are too weak to connect all intra-class points. We compare to the baseline where all augmentations are used.

We use a trained VAE to define $K_{VAE}$ as before and a trained DCGAN (48) $K_{GAN}(\bar{x}, \bar{x}') \stackrel{\text{def}}{=} K(z(\bar{x}), z(\bar{x}'))$ where $z(\cdot)$ denotes the discriminator output of the penultimate layer.

In Table 3, we observe that our contrastive framework with DCGAN representation as prior is able to approach the performance of self-supervised models by applying only crop augmentations and flip. Additionally, when removing almost all augmentations (crop and color distortion), we approach the performance of the prior representations of the generative models. This is expected by our theory

since we have an augmentation graph that is almost disjoint for all points and thus we only rely on the prior to reconnect them. This experiment shows that our method is less sensitive than all other SOTA self-supervised methods to the choice of the "optimal" augmentations, which could be relevant in applications where they are not known a priori, or hard to find.

**Weakly supervised learning on natural images.** In Table 4, we suppose that we have access to image attributes that correlate with the true semantic labels (e.g birds color/size for birds classification). We use three datasets: CUB-200-2011 (59), ImageNet100 (54) and UTZappos (62), following (55). CUB-200-2011 contains 11788 images of 200 bird species with 312 binary attributes available (encoding size, color, etc.). UTZappos contains 50025 images of shoes from several brands sub-categorized into 21 groups that we use as downstream classifica-

| Model | CUB | ImageNet100 | UT-Zappos |
|---|---|---|---|
| SimCLR | 17.48 | 65.30 | 84.08 |
| BYOL | 16.82 | 72.20 | 85.48 |
| CosKernel CCLK (55) | 15.61 | 74.34 | 83.23 |
| RBFKernel CCLK (55) | 30.49 | 77.24 | 84.65 |
| CosKernel Decoupled Unif | 27.77 | **78.8** | **85.56** |
| RBFKernel Decoupled Unif | **32.87** | 76.34 | 84.78 |

Table 4: If images attributes are accessible (e.g birds color or size for CUB200), they can be leveraged as prior in our framework to improve the representation.

tion labels. It comes with seven attributes. Finally, for ImageNet100 we follow (55) and use the pre-trained CLIP (47) model (trained on pairs (text, image)) to extract 512-d features considered as prior information. We compare our method with SOTA Siamese models (SimCLR and BYOL) and with CCLK, a conditional contrastive model that defines positive samples only according to the conditioning attributes. The proposed method outperforms all other models on the three datasets.

**Filling the gap for medical imaging.** Data augmentations on natural images have been handcrafted over decades. However, we argue they are not adapted in other visual domain such as medical imaging (20). We study 1) bipolar disorder detection (BD), a challenging binary classification task, on brain MRI dataset BIOBD (32) and 2) chest radiography interpretation, a 5-class classification task on CheXpert (35). BIOBD contains 356 healthy controls (HC) and 306 patients with BD. We use BHB (20) as a large pre-training dataset containing 10k 3D images of healthy subjects. For CheXpert, we use Gloria (34) representation, a multi-modal approach trained with (medical report, image) pairs to extract 2048-d features as weak annotations. We show that our approach improve contrastive model in both unsupervised (BD) and weakly supervised (CheXpert) setting for medical imaging.

| Model | Atelectasis | Cardiomegaly | Consolidation | Edema | Pleural Effusion |
|---|---|---|---|---|---|
| SimCLR (10) | 82.42 | 77.62 | 90.52 | 89.08 | 86.83 |
| BYOL (26) | 83.04 | 81.54 | 90.98 | 90.18 | 85.99 |
| MoCo-CXR (53) | 75.8 | 73.7 | 77.1 | 86.7 | 85.0 |
| GLoRIA (34) | 86.70 | **86.39** | 90.41 | 90.58 | 91.82 |
| CCLK (55) | 86.31 | 83.67 | 92.45 | 91.59 | 91.23 |
| $K_{GloRIA}$ Decoupled Unif (ours) | **86.92** | 85.88 | **93.03** | **92.39** | **91.93** |
| Supervised (6) | 81.6 | 79.7 | 90.5 | 86.8 | 89.9 |

| Model | BD vs HC |
|---|---|
| SimCLR (10) | $60.46_{\pm1.23}$ |
| BYOL (26) | $58.81_{\pm0.91}$ |
| MoCo v2 (29) | $59.27_{\pm1.50}$ |
| Model Genesis (67) | $59.94_{\pm0.81}$ |
| VAE | $52.86_{\pm1.24}$ |
| $K_{VAE}$ Decoupled Unif (ours) | $\mathbf{62.19}_{\pm1.58}$ |
| Supervised | $67.42_{\pm0.31}$ |

Table 5: AUC scores(%) under linear evaluation for discriminating 5 pathologies on CheXpert. ResNet18 backbone is trained for 400 epochs (batch size $N = 1024$) without labels on official CheXpert training set and results are reported on validation set.

Table 6: BD detection. Linear evaluation AUC scores(%) using a 5-fold leave-site-out CV scheme.

## 5 CONCLUSION

In this work, we have showed that we can integrate prior information into CL to improve the final representation. In particular, we draw connections between kernel theory and CL to build our theoretical framework. We demonstrate tight bounds on downstream classification performance with weaker assumptions than previous works. Empirically, we show that generative models provide a good prior when augmentations are too weak or insufficient to remove easy-to-learn noisy features. We also show applications in medical imaging in both unsupervised and weakly supervised setting where our method outperforms all other models. Thanks to our theoretical framework, we hope that CL will benefit from the future progress in generative modelling and it will widen its field of application to challenging tasks, such as computer aided-diagnosis.

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

# A    MORE EMPIRICAL EVIDENCE

In this section, we provide additional empirical evidence to confirm several claims and arguments developed in the paper.

## A.1    DECOUPLED UNIFORMITY OPTIMIZES ALIGNMENT

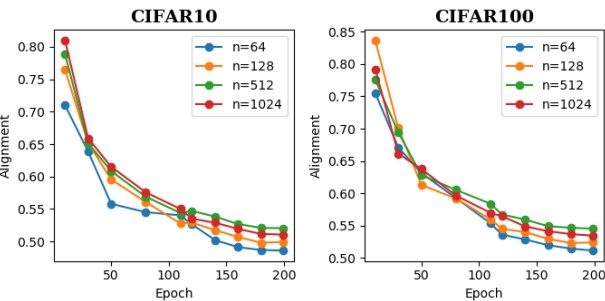

Figure 2: Alignment metric $\mathcal{L}_{align}$ computed on the validation set during optimization of Decoupled Uniformity loss with various batch sizes $n$ and a fixed latent space dimension $d = 128$. We use 100 positive samples per image to compute $\mathcal{L}_{align}$.

We empirically show here that Decoupled Uniformity optimizes alignment, even in the regime when the batch size $n > d + 1$, where $d$ is the representation space dimension. We use CIFAR-10 and CIFAR-100 datasets and we optimize Decoupled Uniformity (without kernel) with all SimCLR augmentations with $d = 128$ and we vary the batch size $n$. We report the alignment metric defined in (57) as $\mathcal{L}_{align} = \mathbb{E}_{\mathcal{A}(x|\bar{x})\mathcal{A}(x'|\bar{x})p(\bar{x})}||f(x) - f(x')||^2$.

## A.2    MEASURING KERNEL QUALITY AND EMPIRICAL VERIFICATION OF OUR THEORY

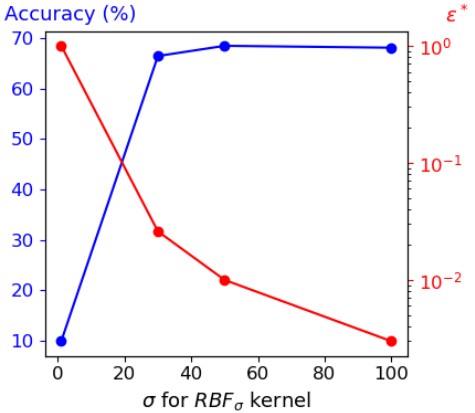 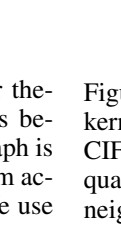 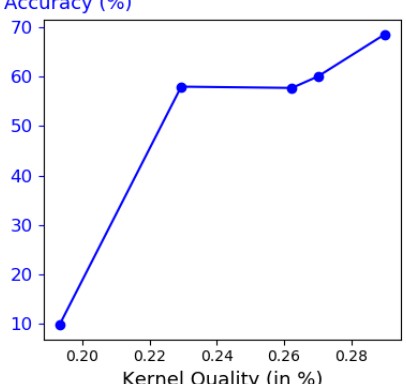

Figure 3: Empirical verification of our theory. The optimal $\epsilon^*$ to add 100 edges between intra-class images in $\epsilon$-Kernel graph is inversely correlated with the downstream accuracy, as suggested by Theorem 4. We use $k = 20$ bits and an RBF kernel.

Figure 4: How we can select *a priori* a good kernel? Downstream accuracy on RandBits CIFAR-10 is highly correlated with kernel quality measured as fraction of 10 nearest neighbors of the same CIFAR-10 class (from test set) in the kernel graph.

We provide empirical evidence confirming our theory (Theorem 4 in particular) along with a new way to quantify kernel quality with respect to a downstream task for a kernel $K$. We perform experiments on RandBits dataset (based on CIFAR-10) with $k = 20$ random bits (almost all points are disconnected in the augmentation graph) and SimCLR augmentations. For a given kernel $K_\sigma$ defined by $K_\sigma(\bar{x}, \bar{x}') = RBF_\sigma(\mu(\bar{x}), \mu(\bar{x}'))$-where $\mu(\cdot)$ is the mean Gaussian distribution of $\bar{x}$ in

VAE latent space trained on RandBits- we train Kernel Decoupled Uniformity with $K_\sigma$ on RandBits. In Fig. 3, we vary $\sigma$ and we report downstream accuracy (measured by linear evaluation) along with the optimal $\epsilon^*$ to add 100 intra-class edges in the $\epsilon$-Kernel graph obtained with $K_\sigma$. The lower $\epsilon^*$, the better the downstream accuracy, which is expected since the upper bound of supervised risk becomes tighter in Theorem 4. It gives a first empirical confirmation that $\epsilon$ tightly bounds the supervised risk on downstream task.

**A new way to quantify kernel quality.** Based on the concept of kernel graph, we measure the quality of a given kernel $K$ using the nearest-neighbors of each image (a vertex in kernel graph). More precisely, $K$ induces a distance $d_K$ ($d_K(a,b) = K(a,a) + K(b,b) - 2K(a,b)$) that can be used to define nearest-neighbors in its kernel graph. We compute the fraction of these nearest neighbors that belong to the same class. In Fig. 4, we plot the downstream accuracy vs kernel quality using 10-nearest neighbors for various kernel $K$. They are obtained by using latent space of a VAE trained for an increasing number of epochs (2, 50, 100, 150 and 1000) and by setting $K(\bar{x}, \bar{x}') = RBF_\sigma(\mu(\bar{x}), \mu(\bar{x}'))$ as before (with $\sigma = 50$ fixed). It shows that this new measure of kernel quality is highly correlated with final downstream accuracy. Therefore, it can be used as a tool to compare *a priori* (without training) different kernels. One limitation of this metric is that it requires access to labels on the downstream task. Future work would consist in finding unsupervised properties of the kernel graph that correlates well with downstream accuracy (e.g. sparsity, clustering coefficient, etc.).

## A.3    MULTI-VIEW CONTRASTIVE LEARNING WITH DECOUPLED UNIFORMITY

When the intra-class connectivity hypothesis is full-filled, we showed that Decoupled Uniformity loss can tightly bound the classification risk for well-aligned encoders (see Theorem 2). Under that hypothesis, we consider the standard empirical estimator of $\mu_{\bar{x}} \approx \sum_{v=1}^{V} f(x^{(v)})$ for $V$ views. Using all SimCLR augmentations, we empirically verify that increasing $V$ allows for: 1) a better estimate of $\mu_{\bar{x}}$ which implies a faster convergence and 2) better SOTA results on both small-scale (CIFAR10, CIFAR100, STL10) and large-scale (ImageNet100) vision datasets. We always use batch size $n = 256$ for all approaches with ResNet18 backbone for CIFAR10, CIFAR100 and STL10 and ResNet50 for ImageNet100. We report the results in Table 7.

| Model | CIFAR-10 | | CIFAR-100 | | ImageNet100 | | STL10 | |
|---|---|---|---|---|---|---|---|---|
| | $e = 200$ | $e = 400$ | $e = 200$ | $e = 400$ | $e = 200$ | $e = 400$ | $e = 200$ | $e = 400$ |
| SimCLR(10) | 79.4 | 81.75 | 48.89 | 53.02 | 65.30 | 66.52 | 76.99 | 79.02 |
| BYOL(26) | 80.14 | 81.97 | 51.57 | 53.65 | 72.20 | 72.26 | 77.62 | 79.61 |
| Decoupled Unif (2 views) | 82.43 | **85.82** | 54.01 | 58.89 | 71.98 | 72.24 | 78.12 | 79.89 |
| Decoupled Unif (4 views) | 84.99 | 85.34 | 57.23 | 59.07 | 72.08 | 75.00 | 78.25 | **80.47** |
| Decoupled Unif (8 views) | **86.50** | 85.80 | **59.63** | **59.74** | **74.70** | **75.00** | **79.82** | 80.30 |

Table 7: A better approximation of centroids $\mu_{\bar{x}}$ (i.e. increasing number of views) when augmentation overlap hypothesis is (nearly) full-filled implies faster convergence. All models are pretrained with batch size $n = 256$. We use ResNet18 backbone for CIFAR10, CIFAR100, STL10 and ResNet50 for ImageNet100. We report linear evaluation accuracy (%) for a given number of epochs $e$.

## A.4    INFLUENCE OF TEMPERATURE AND BATCH SIZE FOR DECOUPLED UNIFORMITY

InfoNCE is known to be sensitive to batch size and temperature to provide SOTA results. In our theoretical framework, we assumed that $f(x) \in \mathbb{S}^{d-1}$ but we can easily extend it to $f(x) \in \sqrt{t}\mathbb{S}^{d-1}$ where $t > 0$ is a hyper-parameter. It corresponds to write $\mathcal{L}^d_{unif}(f) = \mathbb{E}_{p(\bar{x})p(\bar{x}')} e^{-t||\mu_{\bar{x}} - \mu_{\bar{x}'}||^2}$. We show here that Decoupled Uniformity does not require very large batch size (as it is the case for SimCLR) and produce good representations for $t \in [1, 5]$.

| Datasets | $t = 0.1$ | $t = 0.5$ | $t = 1$ | $t = 2$ | $t = 5$ | $t = 10$ |
|----------|-----------|-----------|---------|---------|---------|----------|
| CIFAR10  | 73.91     | 83.01     | 84.72   | **85.82** | 83.05 | 74.82    |
| CIFAR100 | 39.16     | 51.33     | 55.91   | **58.89** | 56.70 | 48.29    |

Table 8: Linear evaluation accuracy (%) after training for 400 epochs with batch size $n = 256$ and varying temperature in Decoupled Uniformity loss with SimCLR augmentations. $t = 2$ gives overall the best results, similarly to the uniformity loss in (57)

| Datasets | Loss | $n = 128$ | $n = 512$ | $n = 1024$ | $n = 2048$ |
|----------|------|-----------|-----------|------------|------------|
| CIFAR10  | SimCLR         | 78.89 | 79.40 | 80.02 | 80.06 |
|          | Decoupled Unif | 82.67 | 82.12 | 82.74 | 82.33 |
| CIFAR100 | SimCLR         | 49.53 | 53.46 | 54.45 | 55.32 |
|          | Decoupled Unif | 54.61 | 54.12 | 55.56 | 55.20 |

Table 9: Linear evaluation accuracy (%) after training for 200 epochs with a batch size $n$, ResNet18 backbone and latent dimension $d = 128$. Decoupled Uniformity is less sensitive to batch size than SimCLR thanks to its decoupling between positives and negatives, similarly to (60).

## A.5 IMPORTANCE OF REGULARIZATION TERM IN CENTROIDS ESTIMATION

Kernel Decoupled Uniformity introduces an additional hyper-parameter $\lambda$ for centroid estimation, which should be such that $\lambda = O\left(\frac{1}{\sqrt{n}}\right)$ where $n$ is the batch size to full-fill the hypothesis of Theorem 4. We have cross-validated this hyper-parameter $\lambda$ on RandBits CIFAR-10 with $k = 10$ bits and we show in Table 10 that $\lambda = \frac{0.01}{\sqrt{n}}$ yields the best results. We have fixed this value for all our experiments in this study.

| $\sqrt{256} \times \lambda$ | $\sigma = 30$ | $\sigma = 50$ |
|-----------------------------|---------------|---------------|
| 0.001 | 10.25     | 60.75     |
| 0.01  | **67.21** | **68.42** |
| 0.1   | 59.09     | 58.13     |
| 1     | 50.49     | 60.75     |

Table 10: Importance of $\lambda$ in centroids estimation with Kernel Decoupled Uniformity. We report linear evaluation accuracy after training on RandBits-CIFAR10 (10 bits) with ResNet18 for 200 epochs using RBFKernel($\sigma$) and batch size $n = 256$.

## A.6 KERNEL CHOICE ON RANDBITS EXPERIMENT

In our experiments on RandBits, we used RBF Kernel in Decoupled Uniformity but other kernels can be considered. Here, we have compared our approach with a cosine kernel on Randbits with $k = 10$ and $k = 20$ bits. There is no hyper-parameter to tune with cosine. From Table 11, we see that cosine gives comparable results for $k = 10$ bits with RBF but it is not appropriate for $k = 20$ bits.

| Kernel | 10 bits | 20 bits |
|--------|---------|---------|
| RBFKernel($\sigma = 1$)  | $66.25_{\pm 0.17}$ | $9.91_{\pm 0.13}$ |
| RBFKernel($\sigma = 30$) | $67.21_{\pm 0.29}$ | $66.46_{\pm 0.19}$ |
| RBFKernel($\sigma = 50$) | $\mathbf{68.42_{\pm 0.51}}$ | $\mathbf{68.58_{\pm 0.17}}$ |
| CosineKernel             | $66.56_{\pm 0.45}$ | $9.68_{\pm 0.18}$ |

Table 11: Linear evaluation after training on RandBits-CIFAR10 with ResNet18 for 200 epochs. RBF and Cosine kernels are evaluated.

## A.7 LARGER PRE-TRAINED GENERATIVE MODEL INDUCES BETTER PRIOR

We argue that using larger datasets (*e.g.,* ImageNet 1K) for pre-training larger generative models will improve the prior on smaller-scale datasets and improve even more the final representations with our method. We have tested this hypothesis on CIFAR-10 and BigBiGAN as prior, compared to DCGAN and the other approaches without prior.

| Model | epochs | CIFAR-10 |
|---|---|---|
| SimCLR (10) | 400 | 81.75 |
| BYOL (26) | 400 | 81.97 |
| MoCov3 (14) | 400 | 86.51 |
| Decoupled Unif | 200 | 82.43 |
| Decoupled Unif | 400 | 85.82 |
| $K_{DCGAN}$ Decoupled Unif | 400 | 85.85 |
| $K_{BigBiGAN}$ Decoupled Unif | 200 | 84.93 (+2.5) |
| $K_{BigBiGAN}$ Decoupled Unif | 400 | **86.86** (+1.04) |

Table 12: We evaluate Kernel Decoupled Uniformity with BigBiGAN pre-trained on ImageNet as prior knowledge. We compare this approach with a shallow DCGAN pre-trained on CIFAR-10 as prior. We train ResNet18 on CIFAR10 and we report linear evaluation accuracy. Pre-trained generative models on larger datasets improve the final representation.

## B GEOMETRICAL CONSIDERATIONS ABOUT DECOUPLED UNIFORMITY

### B.1 ASYMPTOTICAL OPTIMALITY

The assumption $n \leq d + 1$ is crucial to have the existence of a regular simplex on the hypersphere $\mathbb{S}^{d-1}$. In practice, this condition is not always full-filled (e.g SimCLR (10) with $d = 128$ and $n = 4096$). Characterizing the optimal solution of $\mathcal{L}_{unif}^d$ for any $n > d + 1$ is still an open problem (5) but theoretical guarantees can be obtained in the limit case $n \to \infty$.

**Theorem 5.** (Asymptotical Optimality) When the number of samples is infinite $n \to \infty$, then for any perfectly aligned encoder $f \in \mathcal{F}$ that minimizes $\mathcal{L}_{unif}^d$, the centroids $\mu_{\bar{x}}$ for $\bar{x} \sim p(\bar{x})$ are uniformly distributed on the hypersphere $\mathbb{S}^{d-1}$. Proof in Appendix E.2.

Empirically, we observe that minimizers $f$ of $\hat{\mathcal{L}}_{unif}^d$ remain well-aligned when $n > d + 1$ on real-world vision datasets (see Appendix A.1). Decoupled uniformity thus optimizes two properties that are nicely correlated with downstream classification performance (57)–that is alignment and uniformity between centroids. However, as noted in (58; 50), optimizing these two properties is necessary but not sufficient to guarantee a good classification accuracy. In fact, the accuracy can be arbitrarily bad even for perfectly aligned and uniform encoders (50).

### B.2 A METRIC LEARNING POINT-OF-VIEW

In this section, we provide a geometrical understanding of Decoupled Uniformity loss from a metric learning point of view. In particular, we consider the Log-Sum-Exp (LSE) operator often used in CL as an approximation of the maximum.

We consider the finite-samples case with $n$ original samples $(\bar{x}_i)_{i \in [1..n]} \overset{iid}{\sim} p(\bar{x})$ and $V$ views $(x_i^{(v)})_{v \in [1..V]} \overset{iid}{\sim} \mathcal{A}(\cdot | \bar{x}_i)$ for each sample $\bar{x}_i$. We make an abuse of notations and set $\mu_i = \frac{1}{V} \sum_{v=1}^{V} f(x_i^{(v)})$. Then we have:

$$\hat{\mathcal{L}}_{unif}^d = \log \frac{1}{n(n-1)} \sum_{i \neq j} \exp\left(-||\mu_i - \mu_j||^2\right)$$

$$= \log \frac{1}{n(n-1)} \sum_{i \neq j} \exp\left(-s_i^+ - s_j^+ + 2s_{ij}^-\right) \quad (5)$$

where $s_i^+ = ||\mu_i||^2 = \frac{1}{V^2} \sum_{v,v'} s(x_i^{(v)}, x_i^{(v')})$, $s_{ij}^- = \frac{1}{V^2} \sum_{v,v'} s(x_i^{(v)}, x_j^{(v')})$ and $s(\cdot, \cdot) = \langle f(\cdot), f(\cdot) \rangle_2$ is viewed as a similarity measure.

From a metric learning point-of-view, we shall see that minimizing Eq. 5 is (almost) equivalent to looking for an encoder $f$ such that the sum of similarities of all views from the same anchor ($s_i^+$ and $s_j^+$) are higher than the sum of similarities between views from different instances ($s_{ij}^-$):

$$s_i^+ + s_j^+ > 2s_{ij}^- + \epsilon \quad \forall i \neq j \tag{6}$$

where $\epsilon$ is a margin that we suppose "very big" (see hereafter). Indeed, this inequality is equivalent to $-\epsilon > 2s_{ij}^- - s_i^+ - s_j^+$ for all $i \neq j$, which can be written as :

$$\arg\min_f \max(-\epsilon, \{2s_{ij}^- - s_i^+ - s_j^+\}_{i,j \in [1..n], j \neq i})$$

This can be transformed into an optimization problem using the LSE (log-sum-exp) approximation of the $\max$ operator:

$$\arg\min_f \log\left( \exp(-\epsilon) + \sum_{i \neq j} \exp\left(-s_i^+ - s_j^+ + 2s_{ij}^-\right) \right)$$

Thus, if we use an infinite margin ($\lim_{\epsilon \to \infty}$) we retrieve exactly our optimization problem with Decoupled Uniformity in Eq.5 (up to an additional constant depending on $n$).

## C  ADDITIONAL GENERAL GUARANTEES ON DOWNSTREAM CLASSIFICATION

### C.1  OPTIMAL CONFIGURATION OF SUPERVISED LOSS

In order to derive guarantees on a downstream classification task $\mathcal{D}$ when optimizing our unsupervised decoupled uniformity loss, we define a supervised loss that measures the risk on a downstream supervised task. We prove in the next section that the minimizers of this loss have the same geometry as the ones minimizing cross-entropy and SupCon (37): a regular simplex on the hyper-sphere (25). More formally, we have:

**Lemma 6.** Let a downstream task $\mathcal{D}$ with $C$ classes. We assume that $C \leq d + 1$ (*i.e.,* a big enough representation space), that all classes are balanced and the realizability of an encoder $f^* = \arg\min_{f \in \mathcal{F}} \mathcal{L}_{sup}(f)$ with $\mathcal{L}_{sup}(f) = \log \mathbb{E}_{y,y' \sim p(y)p(y')} e^{-||\mu_y - \mu_{y'}||^2}$, and $\mu_y = \mathbb{E}_{p(\bar{x}|y)} \mu_{\bar{x}}$. Then the optimal centroids $(\mu_y^*)_{y \in \mathcal{Y}}$ associated to $f^*$ make a regular simplex on the hypersphere $\mathbb{S}^{d-1}$ and they are perfectly linearly separable, i.e $\min_{(w_y)_{y \in \mathcal{Y}} \in \mathbb{R}^d} \mathbb{E}_{(\bar{x},y) \sim \mathcal{D}} \mathbb{1}(w_y \cdot \mu_y^* < 0) = 0$. Proof in the next section.

This property notably implies that we can realize 100% accuracy at optima with linear evaluation (taking the linear classifier $g(\bar{x}) = W^* f^*(\bar{x})$ with $W^* = (\mu_y^*)_{y \in \mathcal{Y}} \in \mathbb{R}^{C \times d}$).

### C.2  GENERAL GUARANTEES OF DECOUPLED UNIFORMITY

In its most general formulation, we tightly bound the previous supervised loss by Decoupled Uniformity loss $\mathcal{L}_{unif}^d$ depending on a variance term of the centroids $\mu_{\bar{x}}$ conditionally to the labels:

**Theorem 7.** (Guarantees for a given downstream task) For any $f \in \mathcal{F}$ and augmentation $\mathcal{A}$ we have:

$$\mathcal{L}_{unif}^d(f) \leq \mathcal{L}_{sup}(f) \leq 2 \sum_{j=1}^d \text{Var}(\mu_{\bar{x}}^j | y) + \mathcal{L}_{unif}^d(f) \leq 4 \mathbb{E}_{p(\bar{x}|y)p(\bar{x}'|y)} ||\mu_{\bar{x}} - \mu_{\bar{x}'}|| + \mathcal{L}_{unif}^d(f) \tag{7}$$

where $\text{Var}(\mu_{\bar{x}}^j | y) = \mathbb{E}_{p(\bar{x}|y)} (\mu_{\bar{x}}^j - \mathbb{E}_{p(\bar{x}'|y)} \mu_{\bar{x}'}^j)^2$, $y = \arg\max_{y' \in \mathcal{Y}} \text{Var}(\mu_{\bar{x}}^j | y')$ and $\mu_{\bar{x}}^j$ is the $j$-th component of $\mu_{\bar{x}} = \mathbb{E}_{\mathcal{A}(x|\bar{x})} f(x)$. Proof in the next section.

Intuitively, it means that we will achieve good accuracy if all centroids $(\mu_{\bar{x}})_{\bar{x} \in \bar{\mathcal{X}}}$ for samples $\bar{x} \in \bar{\mathcal{X}}$ in the same class are not too far. This theorem is very general since we do not require the intra-class connectivity assumption on $\mathcal{A}$; so any $\mathcal{A} \subset \mathcal{A}^*$ can be used.

## D    EXPERIMENTAL DETAILS

Code will be released upon acceptance of the manuscript. We provide a detailed pseudo-code of our algorithm as well as all experimental details to reproduce the experiments run in the manuscript.

### D.1    PSEUDO-CODE

---
**Algorithm 1** Pseudo-code of the algorithm

---
**Require:** Batch of images $(\bar{x}_1, ..., \bar{x}_n) \in \bar{\mathcal{X}}$, augmentation distribution $\mathcal{A}$, temperature $t$, hyper-parameter $\lambda$ for centroid estimation

$\quad K_n \leftarrow (K(\bar{x}_i, \bar{x}_j))_{i,j \in [1..n]}$          $\triangleright$ Compute the kernel matrix

$\quad \alpha \leftarrow (K_n + n\lambda \mathbf{I}_n)^{-1} K_n$        $\triangleright$ Compute weights for centroid estimation

$\quad x_i^{(1)}, ..., x_i^{(V)} \overset{iid}{\sim} \mathcal{A}(\cdot | \bar{x}_i)$         $\triangleright$ Sample $V$ views per image

$\quad F \leftarrow (\frac{1}{V} \sum_{v=1}^{V} f(x_i^{(v)}))_{i \in [1..n]}$     $\triangleright$ Compute the averaged image representations

$\quad \hat{\mu} \leftarrow \alpha F$                $\triangleright$ Centroid estimation

$\quad \hat{\mathcal{L}}_{unif}^d \leftarrow \log \frac{1}{n(n-1)} \sum_{i \neq j} \exp(-t||\hat{\mu}_i - \hat{\mu}_j||^2)$   $\triangleright$ Kernel Decoupled Uniformity loss

$\qquad\qquad$ **return** $\hat{\mathcal{L}}_{unif}^d$

---

### D.2    IMPLEMENTATION IN PYTORCH

We provide a PyTorch implementation of previous pseudo-code in Algorithm 2. It is generalizable to an arbitrary number of views and kernel.

---
**Algorithm 2** Implementation in PyTorch

---
```python
# loader: generator of images
# n: batch size
# n_views: number of views
# d: latent space dimension
# f: encoder (with projection head)
# x: Tensor of shape [n, *]
# aug: augmentation module generating views
# K: kernel defined on image space
# lamb: hyper-parameter to estimate centroids
for x in loader:
    alphas = (K(x, x) + n*lamb*torch.eye(n)).inverse() @ K(x, x)
    x = aug(x, n_views) # shape=[n*n_views, *]
    z = f(x).view([n, n_views, d]) # shape=[n, n_views, d]
    mu = alphas.detach() @ z.mean(dim=1) # shape=[n, d]
    loss = L(mu)
    loss.backward()

def L(mu, t=2):
    return torch.pdist(mu, p=2).pow(2).mul(-t).exp().mean().log()
```

---

### D.3    DATASETS

**CIFAR (41)**    We use the original training/test split with 50000 and 10000 images respectively of size $32 \times 32$.

**STL-10 (16)**    In unsupervised pre-training, we use all labelled+unlabelled images (105000 images) for training and the remaining 8000 for test with size $96 \times 96$. During linear evaluation, we only use the 5000 training labelled images for learning the weights.

**CUB200-2011 (56)** This dataset is composed of 200 fine-grained bird species with 5994 training images and 5794 test images rescaled to $224 \times 224$.

**UTZappos (62)** This dataset is composed of images of shoes from zappos.com. In order to be comparable with the literature on weakly supervised learning, we follow (55) and split it into 35017 training images and 15008 test images resized at $32 \times 32$.

**ImageNet100 (17; 54)** It is a subset of ImageNet containing 100 random classes and introduced in (54). It contains 126689 training images and 5000 testing images rescaled to $224 \times 224$. It notably allows a reasonable computational time since we runt all our experiments on a single server node with 4 V100 GPU.

**BHB (20)** This dataset is composed of 10420 3D brain MRI images of size $121 \times 145 \times 121$ with $1.5mm^3$ spatial resolution. Only healthy subjects are included.

**BIOBD (32)** It is also a brain MRI dataset including 662 3D anatomical images and used for downstream classification. Each 3D volume has size $121 \times 145 \times 121$. It contains 306 patients with bipolar disorder vs 356 healthy controls and we aim at discriminating patients vs controls. It is particularly suited to investigate biomarkers discovery inside the brain (31).

**CheXpert (35)** This dataset is composed of 224 316 chest radiogaphs of 65240 patients. Each radiograph comes with 14 medical obervations. We use the official training set for our experiments, following (34; 35) and we test the models on the hold-out official validation split containing radiographs from 200 patients. For linear evaluation on this dataset, we train 5 linear probes to discriminate 5 pathologies (as binary classification) using only the radiographs with "certain" labels.

### D.4 Contrastive Models

**Architecture.** For all small-scale vision datasets (CIFAR-10 (41), CIFAR-100 (41), STL-10 (16), CUB200-2011 (56) and UT-Zappos (62)) and CheXpert, we used official ResNet18 (30) backbone where we replaced the first $7 \times 7$ convolutional kernel by a smaller $3 \times 3$ kernel and we removed the first max-pooling layer for CIFAR-10, CIFAR-100 and UTZappos. For ImageNet100, we used ResNet50 (30) for stronger baselines as it is common in the literature. For medical images on brain MRI datasets (BHB (20) and BIOBD(32)), we used DenseNet121 (33) as our default backbone encoder, following previous literature on these datasets (20). We use the official

Following (10), we use the representation space after the last average pooling layer with 2048 dimensions to perform linear evaluation and use a 2-layers MLP projection head with batch normalization between each layer for a final latent space with 128 dimensions.

**Kernel choice.** In all experiments with Kernel Decoupled Uniformity, we used an RBF kernel and we cross-validated the hyperparameter $\sigma$ within $\{0.1, 1, 10, 30, 50, 100\}$.

**Batch size.** We always use a default batch size 256 for all experiments on vision datasets and 64 for brain MRI datasets (considering the computational cost with 3D images and since it had little impact on the performance (20)).

**Optimization.** We use SGD optimizer on small-scale vision datasets (CIFAR-10, CIFAR-100, STL-10, CUB200-2011, UT-Zappos) with a base learning rate $0.3 \times$ batch size$/256$ and a cosine scheduler. For ImageNet100, we use a LARS (61) optimizer with learning rate $0.02 \times \sqrt{\text{batch size}}$ and cosine scheduler. In Kernel Decoupled Uniformity loss, we set $\lambda = \frac{0.01}{\sqrt{\text{batch size}}}$ and $t = 2$. For SimCLR, we set the temperature to $\tau = 0.07$ for all datasets following (60). Unless mentioned otherwise, we use 2 views for Decoupled Uniformity (both with and without kernel) and the computational cost remains comparable with standard contrastive models.

**Training epochs.** By default, we train the models for 200 epochs, unless mentioned otherwise for all vision data-sets excepted CUB200-2011 and UTZappos where we train them for 1000 epochs, following (55) and ImageNet100 where we train them for 400 epochs. For medical brain MRI

dataset, we perform pre-training for 50 epochs, as in (20). As for CheXpert, we train all models for 400 epochs.

**Augmentations.** We follow (10) to define our full set of data augmentations for vision datasets including: *RandomResizedCrop* (uniform scale between 0.08 to 1), *RandomHorizontalFlip* and color distorsion (including color jittering and gray-scale). For medical brain MRI dataset, we use cutout covering 25% of the image in each direction ($1/4^3$ of the entire volume), following (20). For CheXpert, we follow (2) and we use *RandomResizedCrop* (uniform scale between 0.08 to 1), *RandomHorizontalFlip*, *RandomRotation* (up to 45 degrees) however we do not apply color jittering as we work with gray-scale images.

### D.4.1 GENERATIVE MODELS AND GLORIA

**Architecture.** For VAE, we use ResNet18 backbone with a completely symmetric decoder using nearest-neighbor interpolation for up-sampling. For DCGAN, we follow the architecture described in (48). We keep the original dimension for CIFAR-10 and CIFAR-100 datasets and we resize the images to $64 \times 64$ for STL-10. For BigBiGAN (19), we use the ResNet50 pre-trained encoder available at `https://tfhub.dev/deepmind/bigbigan-resnet50/1` with BN+CReLU features.

**Training.** For VAE, we use PyTorch-lightning pre-trained model for STL-10 [3] and we optimize VAE for CIFAR-10 and CIFAR-100 for 400 epochs using an initial learning rate $10^{-4}$ and SGD optimizer with a cosine scheduler. For RandBits experiments, the VAE is trained with the same setup as for CIFAR-10/100 on RandBits-CIFAR10. For DCGAN, we optimize it using Adam optimizer (following (48)) and base learning rate $2 \times 10^{-4}$. Importantly, all generative models are trained without data augmentation, providing a fair comparison with other methods.

**GloRIA(34)** GloRIA can encode both image and text through 2 different encoders. It is pre-trained on the official training set of CheXpert, as in our experiments. We use only GloRIA image's encoder (a ResNet18 in practice[4]) to obtain weak labels on CheXpert and we leverage this weak labels with Kernel Decoupled Uniformity loss. In practice, we use an RBF kernel as in our previous experiments.

### D.4.2 LINEAR EVALUATION

For all experiments, we perform linear evaluation by encoding the original training set (without augmentation) and by training a logistic regression on these features. We cross-validate an $\ell_2$ penalty term between $\{0, 1e-2, 1e-3, 1e-4, 1e-5\}$ for training this linear probe for 300 epochs with an initial learning rate 0.1 decayed by 0.1 at each plateau.

## E PROOFS

### E.1 ESTIMATION ERROR WITH EMPIRICAL DECOUPLED UNIFORMITY

**Property 1.** $\hat{\mathcal{L}}^d_{unif}(f)$ fulfills $|\hat{\mathcal{L}}^d_{unif}(f) - \mathcal{L}^d_{unif}(f)| \leq O\left(\frac{1}{\sqrt{n}}\right)$ with a convergence in law.

PROOF. For any $x \in \mathcal{X}$, since $f(x) \in \mathbb{S}^{d-1}$, then $||\mu_{\bar{x}}|| = ||\mathbb{E}_{\mathcal{A}(x|\bar{x})} f(x)|| \leq \mathbb{E}_{\mathcal{A}(x|\bar{x})} ||f(x)|| = 1$. As a result, $e^{-||\mu_{\bar{x}} - \mu_{\bar{x}'}||^2} \in I \stackrel{\text{def}}{=} [e^{-4}, 1]$ for any $\bar{x}, \bar{x}' \in \bar{\mathcal{X}}$. Since $\log$ is $k$-Lipschitz on $I$ then:

$$|\hat{\mathcal{L}}^d_{unif}(f) - \mathcal{L}^d_{unif}(f)| \leq k \left| \frac{1}{n(n-1)} \sum_{i \neq j} e^{-||\mu_{\bar{x}_i} - \mu_{\bar{x}_j}||^2} - \mathbb{E}_{p(\bar{x})p(\bar{x}')} e^{-||\mu_{\bar{x}} - \mu_{\bar{x}'}||^2} \right|$$

---

[3] `https://github.com/PyTorchLightning/pytorch-lightning`
[4] The official model is available here: `https://github.com/marshuang80/gloria`

For a fixed $\bar{x} \in \bar{\mathcal{X}}$, let $g_n(\bar{x}) = \frac{1}{n} \sum_{i=1}^n e^{-||\mu_{\bar{x}} - \mu_{\bar{x}_i}||^2}$ and $g(\bar{x}) = \mathbb{E}_{p(\bar{x}')} e^{-||\mu_{\bar{x}} - \mu_{\bar{x}'}||^2}$. Since $(Z_i)_{i \in [1..n]} = \left( e^{-||\mu_{\bar{x}} - \mu_{\bar{X}_i}||^2} - g(\bar{x}) \right)_{i \in [1..n]}$ are iid with bounded support in $[-2, 2]$ and zero mean then by Berry–Esseen theorem we have $|g_n(\bar{x}) - g(\bar{x})| \leq O(\frac{1}{\sqrt{n}})$. Similarly, $(Z_i')_{i \in [1..n]} = \left( g_n(\bar{X}_i) - \mathbb{E}_{p(\bar{x})} g_n(\bar{x}) \right)$ are iid, bounded in $[-2, 2]$ and with zero mean. So $|\frac{1}{n} \sum_{i=1}^n g_n(\bar{x}_i) - \mathbb{E}_{p(\bar{x})} g_n(\bar{x})| \leq O(\frac{1}{\sqrt{n}})$ by Berry–Esseen theorem. Then we have:

$$
\begin{aligned}
|\hat{\mathcal{L}}_{unif}^d(f) - \mathcal{L}_{unif}^d(f)| &\leq k |\frac{n}{(n-1)n} \sum_{i=1}^n g_n(\bar{x}_i) - \mathbb{E}_{p(\bar{x})} g(\bar{x})| \\
&\leq 2k |\frac{1}{n} \sum_{i=1}^n g_n(\bar{x}_i) - \mathbb{E}_{p(\bar{x})} g_n(\bar{x}) + \mathbb{E}_{p(\bar{x})} g_n(\bar{x}) - \mathbb{E}_{p(\bar{x})} g(\bar{x})| \\
&\leq O(\frac{1}{\sqrt{n}}) + O(\frac{1}{\sqrt{n}}) \leq O(\frac{1}{\sqrt{n}})
\end{aligned}
$$

### E.2 OPTIMALITY OF DECOUPLED UNIFORMITY

**Theorem 1.** (Optimality of Decoupled Uniformity) Given $n$ points $(\bar{x}_i)_{i \in [1..n]}$ such that $n \leq d+1$, the optimal decoupled uniformity loss is reached when:

1. (Perfect uniformity) All centroids $(\mu_i)_{i \in [1..n]} = (\mu_{\bar{x}_i})_{i \in [1..n]}$ make a regular simplex on the hyper-sphere $\mathbb{S}^{d-1}$

2. (Perfect alignment) $f$ is perfectly aligned, i.e $\forall x, x' \overset{iid}{\sim} \mathcal{A}(\cdot | \bar{x}_i), f(x) = f(x')$

PROOF. We will use Jensen's inequality and basic algebra to show these 2 properties. By triangular inequality, we have $||\mu_i|| = ||\mathbb{E}_{x \sim \mathcal{A}(\cdot | \bar{x}_i)} f(x)|| \leq \mathbb{E} ||f(x)|| = 1$ since we assume $f(x) \in \mathbb{S}^d$. So all $(\mu_i)$ are bounded by 1.

Let $\mu = (\mu_i)_{i \in [1..n]}$. We have:

$$
\begin{aligned}
\Gamma(\mu) := \sum_{i,j=1}^n ||\mu_i - \mu_j||^2 &= \sum_{i,j} ||\mu_i||^2 + ||\mu_j||^2 - 2\mu_i \cdot \mu_j \\
&\leq \sum_{i,j} (2 - 2\mu_i \cdot \mu_j) \\
&= 2n^2 - 2||\sum_i \mu_i||^2 \leq 2n^2
\end{aligned}
$$

with equality if and only if $\sum_{i=1}^n \mu_i = 0$ and $\forall i \in [1..n], ||\mu_i|| = 1$. By strict convexity of $u \to e^{-u}$, we have:

$$
\begin{aligned}
\sum_{i \neq j} \exp(-||\mu_i - \mu_j||^2) &\geq n(n-1) \exp \left( -\frac{\Gamma(\mu)}{n(n-1)} \right) \\
&\geq n(n-1) \exp \left( -\frac{2n}{n-1} \right)
\end{aligned}
$$

with equality if and only if all pairwise distance $||\mu_i - \mu_j||$ are equal (equality case in Jensen's inequality for strict convex function), $\sum_{i=1}^n \mu_i = 0$ and $||\mu_i|| = 1$. So all centroids must form a regular $n-1$-simplex inscribed on the hypersphere $\mathbb{S}^{d-1}$ centered at 0.

Finally, since $||\mu_i|| = 1$ then we have equality in the Jensen's inequality $||\mu_i|| = ||\mathbb{E}_{\mathcal{A}(x|\bar{x}_i)} f(x)|| \leq \mathbb{E}_{\mathcal{A}(x|\bar{x}_i)} ||f(x)|| = 1$. Since $|| \cdot ||$ is strictly convex on the hyper-sphere, then $f$ must be constant on $\text{supp} \, \mathcal{A}(\cdot | \bar{x}_i)$, for all $\bar{x}_i$ so $f$ must be perfectly aligned.

**Theorem 5.** (Asymptotical Optimality) When the number of samples is infinite $n \to \infty$, then for any perfectly aligned encoder $f \in \mathcal{F}$ that minimizes $\mathcal{L}^d_{unif}$, the centroids $\mu_{\bar{x}}$ for $\bar{x} \sim p(\bar{x})$ are uniformly distributed on the hypersphere $\mathbb{S}^{d-1}$.

PROOF. Let $f \in \mathcal{F}$ perfectly aligned. Then all centroids $\mu_{\bar{x}} = f(\bar{x})$ lie on the hypersphere $\mathbb{S}^{d-1}$ and we are optimizing:

$$\arg\min_f \mathcal{L}^d_{unif}(f) = \arg\min_f \mathbb{E}_{\bar{x},\bar{x}' \overset{iid}{\sim} p(\bar{x})} e^{-||f(\bar{x})-f(\bar{x}')||^2}$$

So a direct application of Proposition 1. in (57) shows that the uniform distribution on $\mathbb{S}^{d-1}$ is the unique solution to this problem and that all centroids are uniformly distributed on the hyper-sphere.

### E.3 OPTIMALITY OF SUPERVISED LOSS

**Lemma 6.** Let a downstream task $\mathcal{D}$ with $C$ classes. We assume that $C \leq d + 1$ (*i.e.,* a big enough representation space), that all classes are balanced and the realizability of an encoder $f^* = \arg\min_{f \in \mathcal{F}} \mathcal{L}_{sup}(f)$ with $\mathcal{L}_{sup}(f) = \log \mathbb{E}_{y,y' \sim p(y)p(y')} e^{-||\mu_y - \mu_{y'}||^2}$, and $\mu_y = \mathbb{E}_{p(\bar{x}|y)} \mu_{\bar{x}}$. Then the optimal centroids $(\mu_y^*)_{y \in \mathcal{Y}}$ associated to $f^*$ make a regular simplex on the hypersphere $\mathbb{S}^{d-1}$ and they are perfectly linearly separable, i.e $\min_{(w_y)_{y \in \mathcal{Y}} \in \mathbb{R}^d} \mathbb{E}_{(\bar{x},y) \sim \mathcal{D}} \mathbb{1}(w_y \cdot \mu_y^* < 0) = 0$.

PROOF. This proof is very similar to the one in Theorem 1. We first notice that all "labelled" centroids $\mu_y = \mathbb{E}_{p(\bar{x}|y)} \mu_{\bar{x}}$ are bounded by 1 ($||\mu_y|| \leq \mathbb{E}_{p(\bar{x}|y)} \mathbb{E}_{\mathcal{A}(x|\bar{x})} ||f(x)|| = 1$ by Jensen's inequality applied twice). Then, since all classes are balanced, we can re-write the supervised loss as:

$$\mathcal{L}_{sup}(f) = \log \frac{1}{C^2} \sum_{y,y'=1}^C e^{-||\mu_y - \mu_{y'}||^2}$$

We have:

$$\Gamma_{\mathcal{Y}}(\mu) := \sum_{y,y'=1}^C ||\mu_y - \mu_{y'}||^2 = \sum_{y,y'} ||\mu_y||^2 + ||\mu_{y'}||^2 - 2\mu_y \cdot \mu_{y'}$$
$$\leq \sum_{y,y'} (2 - 2\mu_y \cdot \mu_{y'})$$
$$= 2C^2 - 2||\sum_y \mu_y||^2 \leq 2C^2$$

with equality if and only if $\sum_{y=1}^C \mu_y = 0$ and $\forall y \in [1..C], ||\mu_y|| = 1$. By strict convexity of $u \to e^{-u}$, we have:

$$\sum_{y \neq y'} \exp(-||\mu_y - \mu_{y'}||^2) \geq C(C-1) \exp\left(-\frac{\Gamma_{\mathcal{Y}}(\mu)}{C(C-1)}\right)$$
$$\geq C(C-1) \exp\left(-\frac{2C}{C-1}\right)$$

with equality if and only if all pairwise distance $||\mu_y - \mu_{y'}||$ are equal (equality case in Jensen's inequality for strict convex function), $\sum_{y=1}^C \mu_y = 0$ and $||\mu_y|| = 1$. So all centroids must form a regular $C-1$-simplex inscribed on the hypersphere $\mathbb{S}^{d-1}$ centered at 0. Furthermore, since $||\mu_y|| = 1$ then we have equality in the Jensen's inequality $||\mu_y|| = ||\mathbb{E}_{p(\bar{x}|y)\mathcal{A}(x|\bar{x})} f(x)|| \leq \mathbb{E}_{p(\bar{x}|y)\mathcal{A}(x|\bar{x})} ||f(x)|| = 1$ so $f$ must by perfectly aligned for all samples belonging to the same class: $\forall \bar{x}, \bar{x}' \sim p(\cdot|y), f(\bar{x}) = f(\bar{x}')$.

### E.4 GENERALIZATION BOUNDS FOR DECOUPLED UNIFORMITY

**Theorem 7.** (Guarantees for a given downstream task) For any $f \in \mathcal{F}$ and augmentation distribution $\mathcal{A}$, we have:

$$\mathcal{L}_{unif}^d(f) \leq \mathcal{L}_{unif}^{sup}(f) \leq 2\sum_{j=1}^{d} \text{Var}(\mu_{\bar{x}}^j|y) + \mathcal{L}_{unif}^d(f) \leq 4\mathbb{E}_{p(\bar{x}|y)p(\bar{x}'|y)}||\mu_{\bar{x}} - \mu_{\bar{x}'}|| + \mathcal{L}_{unif}^d(f) \quad (8)$$

where $\text{Var}(\mu_{\bar{x}}^j|y) = \mathbb{E}_{p(\bar{x}|y)}(\mu_{\bar{x}}^j - \mathbb{E}_{p(\bar{x}'|y)}\mu_{\bar{x}'}^j)^2$ and $\mu_{\bar{x}}^j$ is the $j$-th component of $\mu_{\bar{x}} = \mathbb{E}_{\mathcal{A}(x|\bar{x})}f(x)$.

PROOF.

**Lower bound.** To derive the lower bound, we apply Jensen's inequality to convex function $u \to e^{-u}$:

$$\exp\mathcal{L}_{unif}^d(f) = \mathbb{E}_{p(\bar{x})p(\bar{x}')}e^{-||\mu_{\bar{x}} - \mu_{\bar{x}'}||^2}$$
$$= \mathbb{E}_{p(\bar{x}|y)p(\bar{x}'|y)p(y)p(y')}e^{-||\mu_{\bar{x}} - \mu_{\bar{x}'}||^2}$$
$$\leq \mathbb{E}_{p(y)p(y')}\exp\left(-\mathbb{E}_{p(\bar{x}|y)p(\bar{x}'|y')}||\mu_{\bar{x}} - \mu_{\bar{x}'}||^2\right)$$

Then, by Jensen's inequality applied to $||.||^2$:

$$\mathbb{E}_{p(\bar{x}|y)p(\bar{x}'|y')}||\mu_{\bar{x}} - \mu_{\bar{x}'}||^2 \overset{(1)}{=} \mathbb{E}_{p(\bar{x}|y)}||\mu_{\bar{x}}||^2 + \mathbb{E}_{p(\bar{x}'|y')}||\mu_{\bar{x}'}||^2 - 2\mu_y \cdot \mu_{y'}$$
$$\geq ||\mathbb{E}_{p(\bar{x}|y)}\mu_{\bar{x}}||^2 + ||\mathbb{E}_{p(\bar{x}'|y')}\mu_{\bar{x}'}||^2 - 2\mu_y \cdot \mu_{y'}$$
$$\overset{(1)}{=} ||\mu_y - \mu_{y'}||^2$$

(1) follows according to the previous lemma. So we can conclude:

$$\exp\mathcal{L}_{unif}^d(f) \leq \mathbb{E}_{p(y)p(y')}\exp(-||\mu_y - \mu_{y'}||^2) = \exp\mathcal{L}_{unif}^{sup}$$

**Upper bound.** For this bound, we will use the following equality (by definition of variance):

$$||\mathbb{E}_{p(\bar{x}|y)}\mu_{\bar{x}}||^2 = ||\mathbb{E}_{p(\bar{x}|y)}\mu_{\bar{x}}||^2 - \mathbb{E}_{p(\bar{x}|y)}||\mu_{\bar{x}}||^2 + \mathbb{E}_{p(\bar{x}|y)}||\mu_{\bar{x}}||^2$$
$$= -\sum_{j=1}^{d}\text{Var}(\mu_{\bar{x}}^j|y) + \mathbb{E}_{p(\bar{x}|y)}||\mu_{\bar{x}}||^2$$

So we start by expending:

$$||\mu_y - \mu_{y'}||^2 = ||\mathbb{E}_{p(\bar{x}'|y')}\mu_{\bar{x}'}||^2 + ||\mathbb{E}_{p(\bar{x}|y)}\mu_{\bar{x}}||^2 - 2\mathbb{E}_{p(\bar{x}|y)p(\bar{x}'|y')}\mu_{\bar{x}} \cdot \mu_{\bar{x}'}$$
$$= \mathbb{E}_{p(\bar{x}|y)}||\mu_{\bar{x}}||^2 + \mathbb{E}_{p(\bar{x}'|y')}||\mu_{\bar{x}'}||^2 - \left(\sum_{j=1}^{d}\text{Var}(\mu_{\bar{x}}^j|y) + \text{Var}(\mu_{\bar{x}'}^j|y)\right) - 2\mathbb{E}_{p(\bar{x}|y)p(\bar{x}'|y')}\mu_{\bar{x}} \cdot \mu_{\bar{x}'}$$
$$= \mathbb{E}_{p(\bar{x}|y)p(\bar{x}'|y')}||\mu_{\bar{x}} - \mu_{\bar{x}'}||^2 - 2\left(\sum_{j=1}^{d}\text{Var}(\mu_{\bar{x}}^j|y)\right)$$

So by applying again Jensen's inequality:

$$\exp\mathcal{L}_{unif}^{sup} = \mathbb{E}_{p(y)p(y')}\exp(-||\mu_y - \mu_{y'}||^2) \leq \mathbb{E}_{p(y)p(y')}\exp\left(-\mathbb{E}_{p(\bar{x}|y)p(\bar{x}'|y')}||\mu_{\bar{x}} - \mu_{\bar{x}'}||^2 + 2\left(\sum_{j=1}^{d}\text{Var}(\mu_{\bar{x}}^j|y)\right)\right)$$
$$\leq \exp2\left(\sum_{j=1}^{d}\text{Var}(\mu_{\bar{x}}^j|y_m)\right)\mathbb{E}_{p(y)p(y')}\exp\left(-\mathbb{E}_{p(\bar{x}|y)p(\bar{x}'|y')}||\mu_{\bar{x}} - \mu_{\bar{x}'}||^2\right)$$
$$= \exp2\left(\sum_{j=1}^{d}\text{Var}(\mu_{\bar{x}}^j|y_m)\right)\exp\mathcal{L}_{unif}^d$$

We set $y_m = \arg\max_{i,y \in [1..d] \times \mathcal{Y}} \mathrm{Var}(\mu_{\bar{x}}^j | y)$ We conclude here by taking the $\log$ on the previous inequality.

**Variance upper bound.** Starting from the definition of conditional variance:

$$\sum_{j=1}^d \mathrm{Var}(\mu_{\bar{x}}^j | y_m) = \mathbb{E}_{p(\bar{x}|y_m)} ||\mu_{\bar{x}}||^2 - ||\mathbb{E}_{p(\bar{x}|y_m)} \mu_{\bar{x}}||^2$$

$$= \mathbb{E}_{p(\bar{x}|y_m)} \left( (||\mu_{\bar{x}}|| - ||\mathbb{E}_{p(\bar{x}|y_m)} \mu_{\bar{x}}||)(||\mu_{\bar{x}}|| + ||\mathbb{E}_{p(\bar{x}|y_m)} \mu_{\bar{x}}||) \right)$$

$$\overset{(1)}{\leq} \mathbb{E}_{p(\bar{x}|y_m)} ||\mu_{\bar{x}} - \mathbb{E}_{p(\bar{x}'|y_m)} \mu_{\bar{x}'}||(||\mu_{\bar{x}}|| + ||\mathbb{E}_{p(\bar{x}|y_m)} \mu_{\bar{x}}||)$$

$$\overset{(2)}{\leq} 2\mathbb{E}_{p(\bar{x}|y_m)} ||\mu_{\bar{x}} - \mathbb{E}_{p(\bar{x}'|y_m)} \mu_{\bar{x}'}||$$

$$\overset{(3)}{\leq} 2\mathbb{E}_{p(\bar{x}|y_m)p(\bar{x}'|y_m)} ||\mu_{\bar{x}} - \mu_{\bar{x}'}||$$

(1) Follows from standard inequality $||a - b|| \geq |||a|| - ||b|||$ (from Cauchy-Schwarz). (2) follows from boundness of $||\mu_{\bar{x}}|| \leq 1$ and Jensen's inequality. (3) is again Jensen's inequality.

### E.5 GENERALIZATION BOUND UNDER INTRA-CLASS CONNECTIVITY ASSUMPTION

**Theorem 2.** Assuming 1, then for any $\epsilon$-weak aligned encoder $f \in \mathcal{F}$:

$$\mathcal{L}_{unif}^d(f) \leq \mathcal{L}_{unif}^{sup}(f) \leq 8D\epsilon + \mathcal{L}_{unif}^d(f) \tag{9}$$

Where $D$ is the maximum diameter of all intra-class graphs $G_y$ ($y \in \mathcal{Y}$).

PROOF. Let $y \in \mathcal{Y}$ and $\bar{x}, \bar{x}' \sim p(\bar{x}|y)p(\bar{x}'|y)$. By Assumption 1, it exists a path of length $p \leq D$ connecting $(\bar{x}, \bar{x}')$ in $G_y$. So it exists $(\bar{x}_i)_{i \in [1..p+1]} \in \bar{\mathcal{X}}$ and $(x_i)_{i \in [1..p]} \in \mathcal{X}$ s.t $\forall i \in [1..p], x_i \sim \mathcal{A}(x_i|\bar{x}_i) \cap \mathcal{A}(x_i|\bar{x}_{i+1})$, $\bar{x}_1 = \bar{x}$ and $\bar{x}_{p+1} = \bar{x}'$. Then:

$$||\mu_{\bar{x}} - \mu_{\bar{x}'}|| = ||\mu_{\bar{x}_1} - \mu_{\bar{x}_p}||$$

$$= ||\sum_{i=1}^p \mu_{\bar{x}_{i+1}} - \mu_{\bar{x}_i}||$$

$$\leq \sum_{i=1}^p ||\mu_{\bar{x}_{i+1}} - \mu_{\bar{x}_i}||$$

$$= \sum_{i=1}^p ||\mu_{\bar{x}_{i+1}} - f(x_i) + f(x_i) - \mu_{\bar{x}_i}||$$

$$\leq \sum_{i=1}^p ||\mu_{\bar{x}_{i+1}} - f(x_i)|| + ||f(x_i) - \mu_{\bar{x}_i}||$$

$$\overset{(1)}{\leq} \sum_{i=1}^p \mathbb{E}_{p(x|\bar{x}_{i+1})} ||f(x) - f(x_i)|| + \mathbb{E}_{p(x|\bar{x}_i)} ||f(x_i) - f(x)||$$

$$\overset{(2)}{\leq} \sum_{i=1}^p (\epsilon + \epsilon) = 2\epsilon p \leq 2\epsilon D$$

(1) follows from Jensen's inequality and by definition of $\mu_{\bar{x}}$. (2) follows because $f$ is $\epsilon$-weak aligned and $x_i \sim \mathcal{A}(x_i|\bar{x}_i) \cap \mathcal{A}(x_i|\bar{x}_{i+1})$.

So we have $||\mu_{\bar{x}} - \mu_{\bar{x}'}|| \leq 2\epsilon D$ and we can conclude by Theorem 7 (right inequality).

### E.6 CONDITIONAL MEAN EMBEDDING ESTIMATION

Let $f \in \mathcal{F}$ fixed.

**Theorem 3.** (Conditional Mean Embedding estimation) We assume that $\forall g \in \mathcal{H}_{\mathcal{X}}, \mathbb{E}_{p(x|\cdot)} g(x) \in \mathcal{H}_{\bar{\mathcal{X}}}$. Let $\{(x_1, \bar{x}_1), ..., (x_n, \bar{x}_n)\}$ iid samples from $p(x|\bar{x})p(\bar{x})$. Let $\Phi_n = [\phi(\bar{x}_1), ..., \phi(\bar{x}_n)]$ and $\Psi_f = [f(x_1), ..., f(x_n)]^T$. An estimator of the conditional mean embedding is:

$$\forall \bar{x} \in \bar{\mathcal{X}}, \hat{\mu}_{\bar{x}} = \sum_{i=1}^{n} \alpha_i(\bar{x}) f(x_i) \tag{10}$$

where $\alpha_i(\bar{x}) = \sum_{j=1}^{n} [(\Phi_n^T \Phi_n + \lambda n \mathbf{I}_n)^{-1}]_{ij} \langle \phi(\bar{x}_j), \phi(\bar{x}) \rangle_{\mathcal{H}_{\bar{\mathcal{X}}}}$. It converges to $\mu_{\bar{x}}$ with the $\ell_2$ norm at a rate $O(n^{-1/4})$ for $\lambda = O(\frac{1}{\sqrt{n}})$.

PROOF. Let $m_{\bar{x}} = \mathbb{E}_{p(x|\bar{x})} \langle f(x), f(\cdot) \rangle \in \mathcal{H}_{\mathcal{X}}$ be the conditional mean embedding operator. According to Theorem 6 in (52) and the assumption $\forall g \in \mathcal{H}_{\mathcal{X}}, \mathbb{E}_{p(x|\cdot)} g(x) \in \mathcal{H}_{\bar{\mathcal{X}}}$, this estimator can be approximated by:

$$\hat{m}_{\bar{x}} = \sum_{i=1}^{n} \alpha_i(\bar{x}) \langle f(x_i), f(\cdot) \rangle$$

with $\alpha_i$ defined previously in the theorem. This estimator converges with RKHS norm to $m_{\bar{x}}$ at rate $O(\frac{1}{\sqrt{n\lambda}} + \lambda)$. So we need to link $m_{\bar{x}}, \hat{m}_{\bar{x}}$ with $\mu_{\bar{x}}, \hat{\mu}_{\bar{x}}$. We have:

$$\begin{aligned}
\langle m_{\bar{x}}, \hat{m}_{\bar{x}} \rangle_{\mathcal{H}_{\mathcal{X}}} &= \left\langle \mathbb{E}_{p(x|\bar{x})} \langle f(x), f(\cdot) \rangle_{\mathbb{R}^d}, \sum_{i=1}^{n} \alpha_i(\bar{x}) \langle f(x_i), f(\cdot) \rangle_{\mathbb{R}^d} \right\rangle_{\mathcal{H}_{\mathcal{X}}} \\
&= \sum_{i=1}^{n} \alpha_i(\bar{x}) \left\langle \langle \mathbb{E}_{p(x|\bar{x})} f(x), f(\cdot) \rangle_{\mathbb{R}^d}, \langle f(x_i), f(\cdot) \rangle_{\mathbb{R}^d} \right\rangle_{\mathcal{H}_{\mathcal{X}}} \\
&\overset{(1)}{=} \sum_{i=1}^{n} \alpha_i(\bar{x}) \langle \mathbb{E}_{p(x|\bar{x})} f(x), f(x_i) \rangle_{\mathbb{R}^d} \\
&= \langle \mu_{\bar{x}}, \hat{\mu}_{\bar{x}} \rangle_{\mathbb{R}^d}
\end{aligned}$$

(1) holds by the reproducing property of kernel $K_{\mathcal{X}}$ in $\mathcal{H}_{\mathcal{X}}$. We can similarly obtain:

$$\begin{aligned}
||m_{\bar{x}}||_{\mathcal{H}_{\mathcal{X}}}^2 &= \left\langle \mathbb{E}_{p(x|\bar{x})} \langle f(x), f(\cdot) \rangle_{\mathbb{R}^d}, \mathbb{E}_{p(x|\bar{x})} \langle f(x), f(\cdot) \rangle_{\mathbb{R}^d} \right\rangle_{\mathcal{H}_{\mathcal{X}}} \\
&\overset{(1)}{=} \langle \mathbb{E}_{p(x|\bar{x})} f(x), \mathbb{E}_{p(x|\bar{x})} f(x) \rangle_{\mathbb{R}^d} \\
&= ||\mathbb{E}_{p(x|\bar{x})} f(x)||^2 = ||\mu_{\bar{x}}||^2
\end{aligned}$$

Again, (1) by reproducing property of $K_{\mathcal{X}}$. And finally:

$$\begin{aligned}
||\hat{m}_{\bar{x}}||_{\mathcal{H}_{\mathcal{X}}}^2 &= \left\langle \sum_{i=1}^{n} \alpha_i(\bar{x}) \langle f(x_i), f(\cdot) \rangle_{\mathbb{R}^d}, \sum_{i=1}^{n} \alpha_i(\bar{x}) \langle f(x_i), f(\cdot) \rangle_{\mathbb{R}^d} \right\rangle_{\mathcal{H}_{\mathcal{X}}} \\
&= \sum_{i,j} \alpha_i(\bar{x}) \alpha_j(\bar{x}) \langle f(x_i), f(x_j) \rangle_{\mathbb{R}^d} \\
&= ||\hat{\mu}_{\bar{x}}||_{\mathbb{R}^d}^2
\end{aligned}$$

By pooling these 3 equalities, we have:

$$\begin{aligned}
||m_{\bar{x}} - \hat{m}_{\bar{x}}||_{\mathcal{H}_{\mathcal{X}}}^2 &= ||m_{\bar{x}}||^2 + ||\hat{m}_{\bar{x}}||^2 - 2\langle m_{\bar{x}}, \hat{m}_{\bar{x}} \rangle \\
&= ||\mu_{\bar{x}}||^2 + ||\hat{\mu}_{\bar{x}}||^2 - 2\langle \mu_{\bar{x}}, \hat{\mu}_{\bar{x}} \rangle \\
&= ||\mu_{\bar{x}} - \hat{\mu}_{\bar{x}}||_{\mathbb{R}^d}^2
\end{aligned}$$

We can conclude since $||m_{\bar{x}} - \hat{m}_{\bar{x}}|| \leq O(\lambda + (n\lambda)^{-1/2})$.

### E.7 GENERALIZATION BOUND UNDER EXTENDED INTRA-CLASS CONNECTIVITY HYPOTHESIS

**Theorem.** Assuming 3 and 2 holds for a reproducible kernel $K_{\bar{\mathcal{X}}}$ and augmentation distribution $\mathcal{A}$. Let $f \in \mathcal{F}$ $\epsilon'$-aligned. Let $(\bar{x}_i)_{i \in [1..n]}$ be $n$ samples iid drawn from $p(\bar{x})$. We have:

$$\mathcal{L}_{unif}^d(f) \leq \mathcal{L}_{unif}^{sup}(f) \leq \mathcal{L}_{unif}^d(f) + 4D(2\epsilon' + \beta_n(K_{\bar{\mathcal{X}}})\epsilon) + O(n^{-1/4}) \tag{11}$$

where $\beta_n(K_{\bar{\mathcal{X}}}) = (\frac{\lambda_{min}(K_n)}{\sqrt{n}} + \sqrt{n}\lambda)^{-1} = O(1)$ for $\lambda = O(\frac{1}{\sqrt{n}})$, $K_n = (K_{\bar{\mathcal{X}}}(\bar{x}_i, \bar{x}_j))_{i,j \in [1..n]}$ and $D$ is the maximal diameter for all $\tilde{G}_y$, $y \in \mathcal{Y}$. We noted $\lambda_{min}(K_n)$ is the minimal eigenvalue of $K_n$.

PROOF. Let $y \in \mathcal{Y}$ and $\bar{x}, \bar{x}' \sim p(\bar{x}|y)p(\bar{x}'|y)$. By Assumption 2, it exists a path of length $p \leq D$ connecting $\bar{x}, \bar{x}'$ in $\tilde{G}$. So it exists $(\bar{u}_i)_{i \in [1..p+1]} \in \bar{\mathcal{X}}$ and $(u_i)_{i \in I} \in \mathcal{X}$ s.t $\forall i \in I, u_i \sim \mathcal{A}(u_i|\bar{u}_i) \cap \mathcal{A}(u_i|\bar{u}_{i+1})$ and $\forall j \in J, \max(K(\bar{u}_j, \bar{u}_j), K(\bar{u}_{j+1}, \bar{u}_{j+1})) - K(\bar{u}_j, \bar{u}_{j+1}) \leq \epsilon$ with $(I, J)$ a partition of $[1..p]$. Furthermore, $\bar{u}_1 = \bar{x}$ and $\bar{u}_{p+1} = \bar{x}'$. As a result, we have:

$$||\mu_{\bar{x}} - \mu_{\bar{x}'}|| = ||\mu_{\bar{u}_1} - \mu_{\bar{u}_p}||$$

$$= ||\sum_{i=1}^{p} \mu_{\bar{u}_{i+1}} - \mu_{\bar{u}_i}||$$

$$\leq \sum_{i=1}^{p} ||\mu_{\bar{u}_{i+1}} - \mu_{\bar{u}_i}||$$

$$= \sum_{i \in I} ||\mu_{\bar{u}_{i+1}} - \mu_{\bar{u}_i}|| + \sum_{j \in J} ||\mu_{\bar{u}_{j+1}} - \mu_{\bar{u}_j}||$$

**Edges in $E$.** As in proof of Theorem 2, we use the $\epsilon'$-alignment of $f$ to derive a bound:

$$\sum_{i \in I} ||\mu_{\bar{u}_{i+1}} - \mu_{\bar{u}_i}|| = \sum_{i \in I} ||\mu_{\bar{u}_{i+1}} - f(u_i) + f(u_i) - \mu_{\bar{u}_i}||$$

$$\leq \sum_{i \in I} ||\mu_{\bar{u}_{i+1}} - f(u_i)|| + ||f(u_i) - \mu_{\bar{u}_i}||$$

$$\overset{(1)}{\leq} \sum_{i \in I} \mathbb{E}_{p(u|\bar{u}_{i+1})} ||f(u) - f(u_i)|| + \mathbb{E}_{p(u|\bar{u}_i)} ||f(u_i) - f(u)||$$

$$\overset{(2)}{\leq} \sum_{i \in I} (\epsilon' + \epsilon') = 2\epsilon'|I|$$

(1) holds by Jensen's inequality and (2) because $f$ is $\epsilon'$-aligned.

**Edges in $E_K$** For this bound, we will use Theorem 3 to approximate $\mu_{\bar{u}}$ and then derive a bound from the property of $G_K^\epsilon$. Let $(x_k)_{k \in [1.n]} \sim p(x_k|\bar{x}_k)$ $n$ samples iid. By Theorem 3, we know that, for all $j \in J$, $\hat{\mu}_{\bar{u}_j}$ converges to $\mu_{\bar{u}_j}$ with $\ell_2$ norm at rate $O(n^{-1/4})$ where $\hat{\mu}_{\bar{u}_j} = \sum_{k,l=1}^{n} \alpha_{k,l} K_{\bar{\mathcal{X}}}(\bar{x}_l, \bar{u}_j) f(x_k)$ and $\alpha_{k,l} = [(K_n + n\lambda \mathbf{I}_n)^{-1}]_{k,l}$. As a result, for any $j \in J$, we have:

$$||\mu_{\bar{u}_{j+1}} - \mu_{\bar{u}_j}|| = ||\mu_{\bar{u}_{j+1}} - \hat{\mu}_{\bar{u}_{j+1}} + \hat{\mu}_{\bar{u}_{j+1}} - \hat{\mu}_{\bar{u}_j} + \hat{\mu}_{\bar{u}_j} - \mu_{\bar{u}_j}||$$

$$\leq ||\mu_{\bar{u}_{j+1}} - \hat{\mu}_{\bar{u}_{j+1}}|| + ||\hat{\mu}_{\bar{u}_{j+1}} - \hat{\mu}_{\bar{u}_j}|| + ||\hat{\mu}_{\bar{u}_j} - \mu_{\bar{u}_j}|| \overset{(1)}{\leq} O\left(\frac{1}{n^{1/4}}\right) + ||\hat{\mu}_{\bar{u}_{j+1}} - \hat{\mu}_{\bar{u}_j}||$$

Where (1) holds by Theorem 3. Then we will need the following lemma to conclude:

**Lemma.** For any $a, b, c \in \bar{\mathcal{X}}, \max(K(a,a), K(b,b)) - K(a,b) \geq |K(a,c) - K(b,c)|$ for any reproducible kernel K.

PROOF. Let $a, b, c \in \bar{\mathcal{X}}$. We consider the distance $d(x,y) = K(x,x) + K(y,y) - 2K(x,y)$ (it is a distance since $K$ is a reproducible kernel so it can be expressed as $K(\cdot, \cdot) = \langle \phi(\cdot), \phi(\cdot) \rangle$). We will distinguish two cases.

**Case 1.** We assume $K(a,c) \geq K(b,c)$. We have the following triangular inequality:

$$d(a,b) + d(a,c) \geq d(b,c)$$
$$\implies K(a,b) + K(b,b) - 2K(a,b) + K(a,a) + K(c,c) - 2K(a,c) \geq K(b,b) + K(c,c) - 2K(b,c)$$
$$\implies K(a,a) - K(a,b) \geq K(a,c) - K(b,c) \geq 0$$

So $\max(K(a,a), K(b,b)) - K(a,b) \geq |K(a,c) - K(b,c)|$.

**Case 2.** We assume $K(b,c) \geq K(a,c)$. We apply symmetrically the triangular inequality:

$$d(a,b) + d(b,c) \geq d(a,c)$$
$$\implies K(b,b) - K(a,b) \geq K(b,c) - K(a,c) \geq 0$$

So $\max(K(a,a), K(b,b)) - K(a,b) \geq |K(a,c) - K(b,c)|$, concluding the proof.

Then, by definition of $\hat{\mu}_{\bar{u}_j}$:

$$||\hat{\mu}_{\bar{u}_{j+1}} - \hat{\mu}_{\bar{u}_j}|| = ||\sum_{k,l=1}^{n} \alpha_{k,l} K(\bar{x}_l, \bar{u}_{j+1}) f(x_k) - \sum_{k,l=1}^{n} \alpha_{k,l} K(\bar{x}_l, \bar{u}_j) f(x_k)||$$
$$= ||AC||$$

Where $A = (\sum_{k=1}^{n} \alpha_{kj} f(x_k)^i)_{i,j} \in \mathbb{R}^{d \times n}$ ($f(\cdot)^i$ is the i-th component of $f(\cdot)$) and $C = (K(\bar{x}_l, \bar{u}_{j+1}) - K(\bar{x}_l, \bar{u}_j))_l \in \mathbb{R}^{n \times 1}$. So, using the property of spectral $\ell_2$ norm we have:

$$||\hat{\mu}_{\bar{u}_{j+1}} - \hat{\mu}_{\bar{u}_j}|| = ||AC|| \leq ||A||_2 ||C||_2$$

Using the previous lemma and because $(\bar{u}_j, \bar{u}_{j+1}) \in E_K$, we have: $||C||_2^2 = \sum_{i=1}^{n} (K(\bar{x}_i, \bar{u}_{j+1}) - K(\bar{x}_i, \bar{u}_j))^2 \leq \sum_{i=1}^{n} (\max(K(\bar{u}_{j+1}, \bar{u}_{j+1}), K(\bar{u}_j, \bar{u}_j)) - K(\bar{u}_j, \bar{u}_{j+1}))^2 \leq n\epsilon^2$. To conclude, we will prove that $||A||_2 \leq ||\alpha||_2$ where $\alpha = (\alpha_{ij})_{i,j \in [1..n]^2}$. For any $v \in \mathbb{R}^n$, we have:

$$||Av||^2 = ||\sum_{k,j=1}^{n} \alpha_{k,j} v_j f(x_k)||^2 \overset{(1)}{\leq} \left(\sum_{k,j=1}^{n} \alpha_{k,j} v_j\right)^2 = ||\alpha v||^2 \overset{(2)}{\leq} ||\alpha||_2^2 ||v||^2$$

Where (1) holds with Cauchy-Schwarz inequality and because $f(\cdot) \in \mathbb{S}^{d-1}$ and (2) holds by definition of spectral $\ell_2$ norm. So we have $\forall v \in \mathbb{R}^d, ||Av|| \leq ||\alpha||_2 ||v||$, showing that $||A||_2 \leq ||\alpha||_2$.

So we can conclude that:

$$\sum_{j \in J} ||\mu_{\bar{u}_{j+1}} - \mu_{\bar{u}_j}|| \leq \sum_{j \in J} \left(\sqrt{n} ||(K_n + \lambda n \mathbf{I}_n)^{-1}||_2 \epsilon + O(n^{-1/4})\right) = |J| ||(K_n + n\lambda \mathbf{I}_n)^{-1}||_2 \sqrt{n}\epsilon + O(n^{-1/4})$$

We set $\beta_n(K_n) = \sqrt{n} ||(K_n + \lambda n \mathbf{I}_n)^{-1}||_2$. In order to see that $\beta_n(K_n) = (\frac{\lambda_{min}(K_n)}{\sqrt{n}} + \sqrt{n}\lambda)^{-1}$ with $\lambda_{min}(K_n) > 0$ the minimum eigenvalue of $K_n$, we apply the spectral theorem on the symmetric definite-positive kernel matrix $K_n$. Let $0 < \lambda_1 \leq \lambda_2 \leq ... \leq \lambda_n$ the eigenvalues of $K_n$. According to the spectral theorem, it exists $U$ an unitary matrix such that $K_n = UDU^T$ with $D = \text{diag}(\lambda_1, ..., \lambda_n)$. So, by definition of spectral norm:

$$||(K_n + n\lambda \mathbf{I}_n)^{-1}||_2^2 = \lambda_{max} \left(U(D + n\lambda \mathbf{I}_n)^{-1} U^T U (D + \lambda n \mathbf{I}_n)^{-1} U^T\right)$$
$$= \lambda_{max}(U\tilde{D}U^T)$$
$$= (\lambda_1 + n\lambda)^{-2}$$

where $\tilde{D} = \mathrm{diag}(\frac{1}{(\lambda_1 + n\lambda)^2}, ..., \frac{1}{(\lambda_n + n\lambda)^2})$. So we can conclude that $\beta_n(K_n) = (\frac{\lambda_1}{\sqrt{n}} + \sqrt{n}\lambda)^{-1} = O(1)$ for $\lambda = O(\frac{1}{\sqrt{n}})$.

Finally, by pooling inequalities for edges over $E$ and $E_K$, we have:

$$||\mu_{\bar{x}} - \mu_{\bar{x}'}|| \leq 2\epsilon'|I| + |J|\beta_n(K_n)\epsilon + O(n^{-1/4}) \leq D(2\epsilon' + \beta_n(K_n)\epsilon) + O(n^{-1/4})$$

We can conclude by plugging this inequality in Theorem 7.

**Theorem 4.** We assume 2 and 3 hold for a reproducible kernel $K_{\bar{\mathcal{X}}}$ and augmentation distribution $\mathcal{A}$. Let $(x_i, \bar{x}_i)_{i \in [1..n]} \sim \mathcal{A}(x_i, \bar{x}_i)$ iid samples. Let $\hat{\mu}_{\bar{x}_j} = \sum_{i=1}^{n} \alpha_{i,j} f(x_i)$ with $\alpha_{i,j} = ((K_n + \lambda \mathbf{I}_n)^{-1} K_n)_{ij}$ and $K_n = [K_{\bar{\mathcal{X}}}(\bar{x}_i, \bar{x}_j)]_{i,j \in [1..n]}$. Then the empirical decoupled uniformity loss $\hat{\mathcal{L}}_{unif}^d \overset{\text{def}}{=} \log \frac{1}{n(n-1)} \sum_{i,j=1}^{n} \exp(-||\hat{\mu}_{\bar{x}_i} - \hat{\mu}_{\bar{x}_j}||^2)$ verifies, for any $\epsilon'$-weak aligned encoder $f \in \mathcal{F}$:

$$\hat{\mathcal{L}}_{unif}^d - O\left(\frac{1}{n^{1/4}}\right) \leq \mathcal{L}_{unif}^{sup}(f) \leq \hat{\mathcal{L}}_{unif}^d + 4D(2\epsilon' + \beta_n(K_{\bar{\mathcal{X}}})\epsilon) + O\left(\frac{1}{n^{1/4}}\right) \qquad (12)$$

PROOF. We just need to prove that, for any $f \in \mathcal{F}$, $|\mathcal{L}_{unif}^d(f) - \hat{\mathcal{L}}_{unif}^d(f)| \leq O(n^{-1/4})$ and we can conclude through the previous theorem. We have:

$$|\mathcal{L}_{unif}^d(f) - \hat{\mathcal{L}}_{unif}^d(f)| = \left| \log \frac{1}{n(n-1)} \sum_{i,j=1}^{n} \exp(-||\hat{\mu}_{\bar{x}_i} - \hat{\mu}_{\bar{x}_j}||^2) - \mathbb{E}_{p(\bar{x})p(\bar{x}')} e^{-||\mu_{\bar{x}} - \mu_{\bar{x}'}||^2} \right|$$

$$\leq \left| \log \frac{1}{n(n-1)} \sum_{i,j=1}^{n} \exp(-||\hat{\mu}_{\bar{x}_i} - \hat{\mu}_{\bar{x}_j}||^2) - \log \frac{1}{n(n-1)} e^{-||\mu_{\bar{x}_i} - \mu_{\bar{x}_j}||^2} \right|$$

$$+ \left| \log \frac{1}{n(n-1)} e^{-||\mu_{\bar{x}_i} - \mu_{\bar{x}_j}||^2} - \mathbb{E}_{p(\bar{x})p(\bar{x}')} e^{-||\mu_{\bar{x}} - \mu_{\bar{x}'}||^2} \right|$$

The second term in last inequality is bounded by $O(\frac{1}{\sqrt{n}})$ according to property 1. As for the first term, we use the fact that $\log$ is $k$-Lipschitz continuous on $[e^{-4}, 1]$ and $\exp$ is $k'$-Lipschitz continuous on $[-4, 0]$ so:

$$\left| \log \frac{1}{n(n-1)} \sum_{i,j=1}^{n} e^{-||\hat{\mu}_{\bar{x}_i} - \hat{\mu}_{\bar{x}_j}||^2} - \log \frac{1}{n(n-1)} e^{-||\mu_{\bar{x}_i} - \mu_{\bar{x}_j}||^2} \right| \leq \frac{k}{n(n-1)} \left| \sum_{i,j=1}^{n} e^{-||\hat{\mu}_{\bar{x}_i} - \hat{\mu}_{\bar{x}_j}||^2} - e^{-||\mu_{\bar{x}_i} - \mu_{\bar{x}_j}||^2} \right|$$

$$\leq \frac{kk'}{n(n-1)} \left| \sum_{i,j=1}^{n} ||\hat{\mu}_{\bar{x}_i} - \hat{\mu}_{\bar{x}_j}||^2 - ||\mu_{\bar{x}_i} - \mu_{\bar{x}_j}||^2 \right|$$

Finally, we conclude using the boundness of $\hat{\mu}_{\bar{x}}$ and $\mu_{\bar{x}}$ by a constant $C$:

$$||\hat{\mu}_{\bar{x}_i} - \hat{\mu}_{\bar{x}_j}||^2 - ||\mu_{\bar{x}_i} - \mu_{\bar{x}_j}||^2 = (||\hat{\mu}_{\bar{x}_i} - \hat{\mu}_{\bar{x}_j}|| + ||\mu_{\bar{x}_i} - \mu_{\bar{x}_j}||)(||\hat{\mu}_{\bar{x}_i} - \hat{\mu}_{\bar{x}_j}|| - ||\mu_{\bar{x}_i} - \mu_{\bar{x}_j}||)$$
$$\leq 4C(||\hat{\mu}_{\bar{x}_i} - \hat{\mu}_{\bar{x}_j}|| - ||\mu_{\bar{x}_i} - \mu_{\bar{x}_j}||)$$
$$\leq 4C||\hat{\mu}_{\bar{x}_i} - \hat{\mu}_{\bar{x}_j} - (\mu_{\bar{x}_i} - \mu_{\bar{x}_j})||$$
$$\leq 4C(||\hat{\mu}_{\bar{x}_i} - \mu_{\bar{x}_i}|| + ||\hat{\mu}_{\bar{x}_j} - \mu_{\bar{x}_j}||)$$
$$= O\left(\frac{1}{n^{-1/4}}\right)$$

