# OpenReview forum: "Rethinking Positive Sampling for Contrastive Learning with Kernel"
_ICLR.cc/2023/Conference — Submitted to ICLR 2023_

### Official Review · Reviewer_YGqm · 2022-10-15

**Confidence:** 4
**Correctness:** 4
**Technical Novelty And Significance:** 2
**Empirical Novelty And Significance:** 2
**Recommendation:** 5

**Clarity, Quality, Novelty And Reproducibility:**

Clarity and Quality: The paper is well presented. It would be better to change the citation notation from () to [].

Novelty: Positive samples are re-defined through kernel and the prior information can be integrated in the new loss function to further boost the performance.

Reproducibility: The authors have provided details about the training parameters and network architectures in the appendix. These would help to reproduce the experimental results.

**Strength And Weaknesses:**

Strength:

 A new loss function, the decoupled uniformity loss, is provided and further validated theoretically. Prior information is exploited to boost the CL performance.
Both the theoretical and empirical results have been provided regarding the downstream performance.

#################

Weaknesses:

Loss function: The proposed Decoupled Uniformity loss extends the uniformity loss in [54] and does not explicitly include the alignment loss in [54], which is elegant. The authors claimed that the proposed loss function can implicitly encourage alignment (also showed theoretically in Theorem 1 with insufficient training samples). The reviewer wonder what if you add the alignment loss similar to [54]. Would that increase or degrade the performance in practice, ie., when the number of the training samples is neither too low nor infinity? As shown in the experiment, without the prior information, the proposed method cannot beat the SOTA.


Assumptions: The authors introduced the Weak-aligned encoder, which is claimed to be weaker than previous assumptions such as L-smoothness. In the implementation, how to ensure this assumption is satisfied?

Assumptions: How to ensure Assumption 2 in implementation?

Definition 3.7: How to find the value of \lambda?


Experiments:

. From the experimental results, without the prior information (the same as the benchmarks) the proposed method has no advantage compared to the SOTA. The advantage only shows when using the prior knowledge. Such comparison is a bit unfair, because in this case the proposed method essentially requires two representation models learned based on each dataset, ie., VAE/GAN + CL. Such extra complexity and cost need to be considered.

. Kernel quality is crucial to the downstream performance as shown experimentally in Appendix Fig. 4. In the implementation, how to choose a proper kernel with optimal parameters? Are these parameters jointly learned or picked via the validation sets?

. In Table 2, the performance of the proposed method is shown with 4 views. Do the benchmarks also use 4 views? If not, please provide results with 2 views for the fair comparison.

**Summary Of The Paper:**

In the setting of contrastive learning the authors proposed to define positive samples using the kernel theory, and  introduced a new loss function called the decoupled uniformity loss. In addition, the authors suggested to exploit the prior information learned from generative models or given as auxiliary attributes to make it less dependent on data augmentation. Theoretical analysis regarding the downstream classification loss was provided. Experiments were implemented in both unsupervised and weak-supervised settings.

**Summary Of The Review:**

The authors introduced a new loss function, re-defined the positive samples via the kernel, and suggested to boost the performance with prior knowledge. However, as mentioned in the Weaknesses part,

. the introduced assumptions need to be further validated; and

. the proposed method shows no advantage compared to the SOTA under the same condition (ie, without the prior information). When using the prior information, the proposed method essentially needs to learn two representation models based on each dataset. So, the comparison is a bit unfair.

---

### Official Review · Reviewer_6Wk9 · 2022-10-23

**Confidence:** 4
**Correctness:** 3
**Technical Novelty And Significance:** 2
**Empirical Novelty And Significance:** 3
**Recommendation:** 5

**Clarity, Quality, Novelty And Reproducibility:**

It seems that the paper tries to fit many ideas, which might be the reason that I find it a bit hard to navigate.
Here are some notes that may help improve readability.
- When Decoupled Uniformity loss is first introduced, it is unclear to me what's its advantage over SimCLR. It may be helpful to explain that centroids are needed since we can later incorporate the geometry provided by kernels.
- There are no subsections, possibly due to space concerns. I'd suggest keeping subsections for easier reference and better readability, and consider instead rephrasing and reorganizing the contents to save space.
    - For instance, the empirical results on medical imaging currently seems tangent to the rest of the paper, so it may make sense to either move it to the appendix, or restructure the paper to explain why medical domain is key.
    - The part "Geometrical analysis of decoupled uniformity" can also be entirely moved to the appendix.
- Table 3: it's probably more natural to rearrange the columns as "w/o Color and Crop", "w/o Color", "All".
- Typos
    - The paragraph below Theorem 2: "arbitrarily bad".
    - Above Assumption 1: "..that there exists".
    - Extension to multi-views: there's typo in the definition.
        - the LHS of the last equality should be $\hat{\mu}_{\bar{x}_j}^{(v)}$.
    - E.1, property 1: should $k$ be $K$ i.e. the number of classes?
        - Also, should the statement be a high probability statement?
- Please fix the format for the double quotes.


**Strength And Weaknesses:**

Strengths:
- I find the idea of incorporating prior knowledge in form of kernels to be reasonable ("unsurprising"), yet still interesting.
- The empirical results adds one more piece of evidence that leveraging generative models in contrastive representation learning may be promising.

Concerns and questions
- The empirical comparison could be done more thoroughly, and the design choices could be better justified.
  - Table 2: what accuracy does SimCLR achieve?
  - What's the reason for using BigBiGAN for Table 2, but DCGAN for Table 3?
- The paper should cite and compare with "Can contrastive learning avoid shortcut solutions?" (Robinson et al. 2021)
  -  Also in Sec 3, the comment on InfoNCE that the "extension to multiple views is not straightforward": the paper "Revisiting Contrastive Learning through the Lens of Neighborhood Component Analysis: an Integrated Framework" seems related.
- Table 3: the comparison between 1) the gap between "w/o Color" and "All" and 2) the gap between ("w/o Color and Crop" and "w/o Color") seems to vary across setups. Did you find any consistent trend, e.g. a particular type of generative model is more robust to the absence of certain type of augmentation?
- Improvement over generative features may be marginal: the proposed framework can be viewed equivalently as using generative features to improve contrastive learning, or using contrastive learning to improve feature quality of generative modeling. The latter view may be more appropriate sometimes, since the generative features may already achieve competitive accuracy.
    - Table 2: the BigBiGAN features themselves already achieve good accuracy, so the accuracy gain from training on decoupled uniformity loss seems marginal.
    - Table 3, STL-10: when there is no augmentation, DCGAN is better than Decoupled loss+DCGAN (do you know why?).
- How to choose the kernel in practice? Related to the previous point, it seems that if the kernel comes from a generative model, then the quality of the generative model's feature is strongly correlated with the quality of the kernel; can we use contrastive learning and generative model together, either training jointly or iteratively boosting each other?

Clarifications/comments:
- "...we see that Decoupled Uniformity can realize both perfect alignment and uniformity, contrary to InfoNCE": why can't InfoNCE realize both?
- The comments under Thm 3 says it "provides tighter lower bound"; how is Thm 3 a lower bound?


**Summary Of The Paper:**

The success of contrastive learning has been heavily relying on the quality of data augmentations.
This paper tries to reduce this reliance by augmenting the augmentation graph with some other "kernel graph", where the kernel is defined with prior knowledge, such as features of generative models or attributes of data points. The paper also proposes a "decoupled uniformity loss" to allow incorporating such kernel.

The paper theoretically justify that the inclusion of such kernels can relax the assumption on data augmentation, and empirically show that adding such a kernel helps to improve contrastive methods' performance especially on poor augmentation. The quality of the kernel (e.g. features from $\beta$-VAE vs from DCGAN) affects the accuracy as expected.

**Summary Of The Review:**

This paper proposes to use prior information to make contrastive learning more robust to the choice of data augmentation.
I find the overall message interesting, but believe that the paper could be strengthened with more carefully conducted experiments and better organization.

---

### Official Review · Reviewer_khWQ · 2022-10-24

**Confidence:** 4
**Correctness:** 3
**Technical Novelty And Significance:** 3
**Empirical Novelty And Significance:** 3
**Recommendation:** 5

**Clarity, Quality, Novelty And Reproducibility:**

**Clarity**

The paper is overall clear to the readers. I list some points that are not precise below:

Please clarify on which datasets the generative models are pre-trained. For example, in the K-VAE experiment shown in Table 1, is the $\beta$-VAE pre-trained on normal CIFAR10 or Randbits CIFAR10? If we leverage the generative models that are pre-trained on larger datasets, will the performance gets improved?

Considering the training of the generative models also involves augmentations like random crop, random flip, etc., it is unclear whether these augmentations would bring in additional information via the pre-trained generative models when the contrastive learning part does not involve crop and colorization augmentations. The details regarding the training of generative models may also be needed to clarify.



**Novelty and quality**

The proposed method has novelty in improving the positive distribution modeling and shows comprehensive theoretical analysis. However, the empirical study does not well support the proposed method and the effect of the proposed method still remains unclear under several conditions. Compared to previous methods that use simple augmentations, it is hard to convince practitioners to choose the proposed method rather than using augmentations.

**Reproducibility**

The authors include details for the algorithm and model parameterization, which should be useful for reproducing their results.

**Strength And Weaknesses:**

**Strength**

The paper is well-written and organized, making it easy to follow. The proposed loss is simple and is well presented in a mathematical way. Following the theoretical results, some interpretations are provided to help the readers to understand. The authors leverage the kernel graph theory to generalize the positive sample distribution and is empirically justified to be good when crop and coloring augmentation is not applied. From the empirical perspective, the authors have shown their main experiments reasonably, and the details included should be sufficient for reproducibility.

**Weakness**

The paper uses several paragraphs to discuss the weak and strong augmentation, but the definition of weak and strong is heuristic, and it seems the crop and coloring augmentations are empirically regarded as strong augmentations regardless of the dataset and the data distribution.

Although formulated as a general method, the proposed method is still justified on image datasets. For some other data domains like graph, the effectiveness of this method is unknown.

As the method involves a kernel estimation in calculating the centroid, it introduces a new hyperparameter $\lambda$, but I did not find the study regarding the effect of this hyper-parameter in the experiment (please correct me if I missed out this part). Moreover, the choice of the kernel is also a part that needs further tuning in practical usage.

The downstream task is only assumed to be classification, but there could exist various other options. When using generative models, it requires additional training cost if there are no applicable pre-trained models.

From the perspective of empirical studies, the improvements only appear in the case when crop and coloring are not applied; while combining these two augmentations, the proposed method underperforms previous methods.

The baselines of multi-view experiments are relatively weak. If compared with stronger methods that use multiple positives (SWAV, Dino, CACR, F-CL), there is a significant performance gap.

(SWAV: https://arxiv.org/abs/2006.09882
Dino: https://arxiv.org/abs/2104.14294
CACR: https://arxiv.org/abs/2105.03746
F-CL: https://arxiv.org/abs/2011.11765
)

**Summary Of The Paper:**

This paper presents a contrastive learning method that generalizes the previous CL loss in a novel kernel-based prototypical loss uniform loss form, and integrates kernel methods for positive distribution modeling with generative models to show the effectiveness in the case where crop and coloring augmentations are not used. The authors provide theoretical analysis to show the proposed loss is able to bound the downstream classification loss. The authors conduct the empirical studies on both natural images like CIFAR10/100, STL, ImageNet100 and medical images.

**Summary Of The Review:**

The problem pointed out in this paper is important in the study of contrastive learning. The proposed method is also simple and shows improvements in the case when crop and coloring are not applied. However, some key perspectives regarding the proposed method are not comprehensively studied and thus remain unclear to the readers, which weakens the contribution of this paper, and I prefer to vote this paper below the acceptance threshold.

---

### Official Review · Reviewer_rmip · 2022-10-25

**Confidence:** 4
**Correctness:** 4
**Technical Novelty And Significance:** 4
**Empirical Novelty And Significance:** 4
**Recommendation:** 6

**Clarity, Quality, Novelty And Reproducibility:**

The paper is generally clear, although a bit crowded. The ideas are quite clear and presented in a systematic and rigorous manner. The results are new both theoretically and empirically. The paper has enough information for reproducibility.

I would say it is better to expand on the intuition of theorem 5, perhaps at the expense of earlier discussion on uniformity and alignment. Specifically, why should minimum eigenvalue of K matter? and how does that relate to the fact that you do thresholding to obtain the graph?

**Strength And Weaknesses:**

Overall, the paper is quite interesting and is presented in a complete and rigorous manner. The paper develops everything is a very principled way, which is one good feature of this paper. Theorems are insightful, supports intuition and lead to an empirically powerful algorithm. Overall, very nice paper.

The only weakness, perhaps, from my perspective is the following. Contrastive learning is supposed to be a method of representation learning. If we need yet another representation learning method like VAE or GAN, then it looks quite a lot of work just to do classification. I understand that the authors tested their method on downstream task of classification because that is what is shown in the literature. However, it would improve the paper a lot if the authors could show the importance of features learnt from this kind of contrastive learning are far superior to both the VAE  and regular contrastive learning.

**Summary Of The Paper:**

Based on the perspective of Decoupled Uniformity, the authors propose a new loss and new perspective in contrastive learning. Introducing centroid based loss opens up a couple of interesting results like  1) the loss leads to uniformity and alignment (the idea of uniformity and alignment is from earlier paper), 2) the loss could be used to bound the classification loss (related to downstream task) and 3) exploiting the connection to kernel methods and mean embedding, prior information can be incorporated to design better contrastive learning. Using the pretrained VAE and GANs authors show how their contrastive learning is more robust and accurate by exploiting the kernel formulation

**Summary Of The Review:**

The paper is well presented and contains enough high quality results. So, I recommend accept. There are some minor concerns. I hope authors can clarify those.

---

### Decision · Program_Chairs · 2023-01-20

**Decision:**

Reject

**Justification For Why Not Higher Score:**

After the rebuttal period three out of four reviewers vote for rejection of the paper.

**Justification For Why Not Lower Score:**

N/A

**Metareview: Summary, Strengths And Weaknesses:**

Based on the perspective of Decoupled Uniformity, the authors propose a new loss for contrastive learning and and integrate kernel methods for positive distribution modelling.

A weakness of the paper highlighted by the reviewer is that the algorithm relies on strong assumptions that are hard to validate.
It is also unclear how to select and tune a good kernel for the method to work.
The baselines of multi-view experiments are relatively weak.
The experiments to not show if the method generalises beyond images.